# Near-Optimal Comparison Based Clustering

**Michaël Perrot**[*]
Univ Lyon, UJM-Saint-Etienne, CNRS, IOGS,
LabHC UMR 5516, F-42023, SAINT-ETIENNE, France
`michael.perrot@univ-st-etienne.fr`

**Pascal Mattia Esser**[*],     **Debarghya Ghoshdastidar**[*]
Department of Informatics
Technical University of Munich
`{esser,ghoshdas}@in.tum.de`

## Abstract

The goal of clustering is to group similar objects into meaningful partitions. This process is well understood when an explicit similarity measure between the objects is given. However, far less is known when this information is not readily available and, instead, one only observes ordinal comparisons such as *"object i is more similar to j than to k."* In this paper, we tackle this problem using a two-step procedure: we estimate a pairwise similarity matrix from the comparisons before using a clustering method based on semi-definite programming (SDP). We theoretically show that our approach can exactly recover a planted clustering using a near-optimal number of passive comparisons. We empirically validate our theoretical findings and demonstrate the good behaviour of our method on real data.

## 1   Introduction

In clustering, the objective is to group together objects that share the same semantic meaning, that are similar to each other, into $k$ disjoint partitions. This problem has been extensively studied in the literature when a measure of similarity between the objects is readily available, for example when the examples have a Euclidean representation or a graph structure (Shi and Malik, 2000; Arthur and Vassilvitskii, 2007; von Luxburg, 2007). However, it has attracted less attention when the objects are difficult to represent in a standard way, for example cars or food. A recent trend to tackle this problem is to use comparison based learning (Ukkonen, 2017; Emamjomeh-Zadeh and Kempe, 2018) where, instead of similarities, one only observes comparisons between the examples:
**Triplet comparison:** Object $x_i$ is more similar to object $x_j$ than to object $x_k$;
**Quadruplet comparison:** Objects $x_i$ and $x_j$ are more similar to each other than objects $x_k$ and $x_l$.
There are two ways to obtain these comparisons. On the one hand, one can adaptively query them from an oracle, for example a crowd. This is the active setting. On the other hand, they can be directly given, with no way to make new queries. This is the passive setting. In this paper, we study comparison based learning for clustering using passively obtained triplets and quadruplets.

Comparison based learning mainly stems from the psychometric and crowdsourcing literature (Shepard, 1962; Young, 1987; Stewart et al., 2005) where the importance and robustness of collecting ordinal information from human subjects has been widely discussed. In recent years, this framework has attracted an increasing amount of attention in the machine learning community and three main learning paradigms have emerged. The first one consists in obtaining an Euclidean embedding of the data that respects the comparisons as much as possible and then applying standard learning

---

[*]Equal contribution.

techniques (Borg and Groenen, 2005; Agarwal et al., 2007; Jamieson and Nowak, 2011; Tamuz et al., 2011; van der Maaten and Weinberger, 2012; Terada and von Luxburg, 2014; Zhang et al., 2015; Amid and Ukkonen, 2015; Arias-Castro, 2017). The second paradigm is to directly solve a specific task from the ordinal comparisons, such as data dimension or density estimation (Kleindessner and von Luxburg, 2015; Ukkonen et al., 2015), classification and regression (Haghiri et al., 2018), or clustering (Vikram and Dasgupta, 2016; Ukkonen, 2017; Ghoshdastidar et al., 2019). Finally, the third paradigm is an intermediate solution where the idea is to learn a similarity or distance function, as in embedding approaches, but, instead of satisfying the comparisons, the objective is to solve one or several standard problems such as classification or clustering (Kleindessner and von Luxburg, 2017). In this paper, we focus on this third paradigm and propose two new similarities based on triplet and quadruplet comparisons respectively. While these new similarities can be used to solve any machine learning problem, we show that they are provably good for clustering under a well known planted partitioning framework (Abbe, 2017; Yan et al., 2018; Xu et al., 2020).

**Motivation of this work.** A key bottleneck in comparison based learning is the overall number of available comparisons: given $n$ examples, there exist $\mathcal{O}\left(n^3\right)$ different triplets and $\mathcal{O}\left(n^4\right)$ different quadruplets. In practice, it means that, in most applications, obtaining all the comparisons is not realistic. Instead, most approaches try to use as few comparisons as possible. This problem is relatively easy when the comparisons can be actively queried and it is known that $\Omega\left(n \ln n\right)$ adaptively selected comparisons are sufficient for various learning problems (Haghiri et al., 2017; Emamjomeh-Zadeh and Kempe, 2018; Ghoshdastidar et al., 2019). On the other hand, this problem becomes harder when the comparisons are passively obtained. The general conclusion in most theoretical results on learning from passive ordinal comparisons is that, in the worst case, almost all the $\mathcal{O}\left(n^3\right)$ or $\mathcal{O}\left(n^4\right)$ comparisons should be observed (Jamieson and Nowak, 2011; Emamjomeh-Zadeh and Kempe, 2018). The focus of this work is to show that, by carefully handling the passively obtained comparisons, it is possible to design comparison based approaches that use almost as few comparisons as active approaches for planted clustering problems.

**Near-optimal guarantees for clustering with passive comparisons.** In hierarchical clustering, Emamjomeh-Zadeh and Kempe (2018) showed that constructing a hierarchy that satisfies all comparisons in a top-down fashion requires $\Omega\left(n^3\right)$ passively obtained triplets in the worst case. Similarly, Ghoshdastidar et al. (2019) considered a planted model and showed that $\Omega\left(n^{3.5} \ln n\right)$ passive quadruplets suffice to recover the true hierarchy in the data using a bottom-up approach. Since the main difficulty lies in recovering the small clusters at the bottom of the tree, we believe that this latter result also holds for standard clustering. In this paper, we consider a planted model for standard clustering and we show that, when the number of clusters $k$ is constant, $\Omega\left(n(\ln n)^2\right)$ passive triplets or quadruplets are sufficient for exact recovery.[2] This result is comparable to the sufficient number of active comparisons in most problems, that is $\Omega\left(n \ln n\right)$ (Haghiri et al., 2017; Emamjomeh-Zadeh and Kempe, 2018). Furthermore, it is near-optimal. Indeed, to cluster an example, it is necessary to observe it in a comparison at least once as, otherwise, it can only be assigned to a random cluster. Thus, to cluster $n$ objects, it is necessary to have access to at least $\Omega\left(n\right)$ comparisons. Finally, to obtain these results, we study a semi-definite programming (SDP) based clustering method and our analysis could be of significant interest beyond the comparison based framework.

**General noise model for comparison based learning.** In comparison based learning, there are two main sources of noise. First, the observed comparisons can be noisy, that is the observed triplets and quadruplets are not in line with the underlying similarities. This noise stems, for example, from the randomness of the answers gathered from a crowd. It is typically modelled by assuming that each observed comparison is randomly (and independently) flipped (Jain et al., 2016; Emamjomeh-Zadeh and Kempe, 2018). This is mitigated in the active setting by repeatedly querying each comparison, but may have a significant impact in the passive setting where a single instance of each comparison is often observed. Apart from the aforementioned observation errors, the underlying similarities may also have intrinsic noise. For instance, the food data set by Wilber et al. (2014) contains triplet comparisons in terms of which items taste more similar, and it is possible that the taste of a dessert is closer to a main dish than to another dessert. This noise has been considered in Ghoshdastidar et al. (2019) by assuming that every pair of items possesses a latent random similarity, which affects the

responses to comparisons. In this paper, we propose, to the best of our knowledge, the first analysis that considers and shows the impact of both types of noise on the number of passive comparisons.

**Scalable comparison based similarity functions.** Several similarity and kernel functions have been proposed in the literature (Kleindessner and von Luxburg, 2017; Ghoshdastidar et al., 2019). However, computing these similarities is usually expensive as they require up to $\mathcal{O}\left(n\right)$ passes over the set of available comparisons. In this paper, we propose new similarity functions whose construction is much more efficient than previous kernels. Indeed, they can be obtained with a single pass over the set of available comparisons. It means that our similarity functions can be computed in an online fashion where the comparisons are obtained one at a time from a stream. The main drawback compared to existing approaches is that we lose the positive semi-definiteness of the similarity matrix, but our theoretical results show that this is not an issue in the context of clustering. We also demonstrate this empirically as our similarities obtain results that are comparable with state of the art methods.

## 2  Background and theoretical framework

In this section, we present the comparison based framework and our planted clustering model, under which we later show that a small number of passive comparisons suffices for learning. We consider the following setup. There are $n$ items, denoted by $[n] = \{1, 2, \ldots, n\}$, and we assume that, for every pair of distinct items $i, j \in [n]$, there is an implicit real-valued similarity $w_{ij}$ that we cannot directly observe. Instead, we have access to

$$
\begin{aligned}
\text{Triplets:} \quad & \mathcal{T} = \left\{ (i,j,r) \in [n]^3 \ : \ w_{ij} > w_{ir}, \ i,j,r \text{ distinct} \right\}, \qquad \text{or} \\
\text{Quadruplets:} \quad & \mathcal{Q} = \left\{ (i,j,r,s) \in [n]^4 \ : \ w_{ij} > w_{rs}, \ i \neq j, r \neq s, (i,j) \neq (r,s) \right\}.
\end{aligned}
\tag{1}
$$

There are $\mathcal{O}\left(n^4\right)$ possible quadruplets and $\mathcal{O}\left(n^3\right)$ possible triplets, and it is expensive to collect such a large number of comparisons via crowdsourcing. In practice, $\mathcal{T}$ or $\mathcal{Q}$ only contain a small fraction of all possible comparisons. We note that if a triple $i, j, r \in [n]$ is observed with $i$ as reference item, then either $(i,j,r) \in \mathcal{T}$ or $(i,r,j) \in \mathcal{T}$ depending on whether $i$ is more similar to $j$ or to $r$. Similarly, when tuples $(i,j)$ and $(r,s)$ are compared, we have either $(i,j,r,s) \in \mathcal{Q}$ or $(r,s,i,j) \in \mathcal{Q}$.

**Sampling and noise in comparisons.** This paper focuses on passive observation of comparisons. To model this, we assume that the comparisons are obtained via uniform sampling, and every comparison is equally likely to be observed. Let $p \in (0,1]$ denote a sampling rate that depends on $n$. We assume that every comparison (triplet or quadruplet) is independently observed with probability $p$. In expectation, $|\mathcal{Q}| = \mathcal{O}\left(pn^4\right)$ and $|\mathcal{T}| = \mathcal{O}\left(pn^3\right)$, and we can control the sampling rate $p$ to study the effect of the number of observations, $|\mathcal{Q}|$ or $|\mathcal{T}|$, on the performance of an algorithm.

As noted in the introduction, the observed comparisons are typically noisy due to random flipping of answers by the crowd workers and inherent noise in the similarities. To model the external (crowd) noise we follow the work of Jain et al. (2016) and, given a parameter $\epsilon \in (0,1]$, we assume that any observed comparison is correct with probability $\frac{1}{2}(1 + \epsilon)$ and flipped with probability $\frac{1}{2}(1 - \epsilon)$. To be precise, for observed triple $i, j, r \in [n]$ such that $w_{ij} > w_{ir}$,

$$
\mathbf{P}\big((i,j,r) \in \mathcal{T} \mid w_{ij} > w_{ir}\big) = \frac{1 + \epsilon}{2}, \quad \text{whereas} \quad \mathbf{P}\big((i,r,j) \in \mathcal{T} \mid w_{ij} > w_{ir}\big) = \frac{1 - \epsilon}{2}.
\tag{2}
$$

The probabilities for flipping quadruplets can be similarly expressed. We model the inherent noise by assuming $w_{ij}$ to be random, and present a model for the similarities under planted clustering.

**Planted clustering model.** We now present a theoretical model for the inherent noise in the similarities that reflects a clustered structure of the items. The following model is a variant of the popular stochastic block model, studied in the context of graph clustering (Abbe, 2017), and is related to the non-parametric weighted stochastic block model (Xu et al., 2020).

We assume that the item set $[n]$ is partitioned into $k$ clusters $\mathcal{C}_1, \ldots, \mathcal{C}_k$ of sizes $n_1, \ldots, n_k$, respectively, but **the number of clusters $k$ as well as the clusters $\mathcal{C}_1, \ldots, \mathcal{C}_k$ are unknown to the algorithm.** Let $F_{in}$ and $F_{out}$ be two distributions defined on $\mathbb{R}$. We assume that the inherent (and unobserved) similarities $\{w_{ij} : i < j\}$ are random and mutually independent, and

$$
w_{ij} \sim F_{in} \quad \text{if } i,j \in C_\ell \text{ for some } \ell, \qquad \text{and} \qquad w_{ij} \sim F_{out} \quad \text{otherwise.}
$$

We further assume that $w_{ii}$ is undefined, $w_{ji} = w_{ij}$, and that for $w, w'$ independent,

$$
\begin{aligned}
\mathbf{P}_{w,w' \sim F_{in}}(w > w') &= \mathbf{P}_{w,w' \sim F_{out}}(w > w') = 1/2, \quad \text{and} \\
\mathbf{P}_{w \sim F_{in}, w' \sim F_{out}}(w > w') &= (1 + \delta)/2 \quad \text{for some } \delta \in (0, 1].
\end{aligned}
\tag{3}
$$

The first condition in (3) requires that $F_{in}, F_{out}$ do not have point masses, and is assumed for analytical convenience. The second condition ensures that within cluster similarities are larger than inter-cluster similarities—a natural requirement. Ghoshdastidar et al. (2019) used a special case of the above model, where $F_{in}, F_{out}$ are assumed to be Gaussian with identical variances $\sigma^2$, and means satisfy $\mu_{in} > \mu_{out}$. In this case, $\delta = 2\Phi\big((\mu_{in} - \mu_{out})/\sqrt{2}\sigma\big) - 1$ where $\Phi$ is the cumulative distribution function of the standard normal distribution.

**The goal of this paper is to obtain bounds on the number of passively obtained triplets/quadruplets that are sufficient to recover the aforementioned planted clusters with zero error.** To this end, we propose two similarity functions respectively computed from triplet and quadruplet comparisons, and show that a similarity based clustering approach using semi-definite programming (SDP) can exactly recover clusters planted in the data using few passive comparisons.

## 3 A theoretical analysis of similarity based clustering

Before presenting our new comparison based similarity functions, we describe the SDP approach for clustering from similarity matrices that we use throughout the paper (Yan et al., 2018; Chen and Yang, 2020). In addition, we prove a generic theoretical guarantee for this approach that holds for any similarity matrix and, thus, that could be of interest even beyond the comparison based setting.

Similarity based clustering is widely used in machine learning, and there exist a range of popular approaches including spectral methods (von Luxburg, 2007), semi-definite relaxations (Yan and Sarkar, 2016), or linkage algorithms (Dasgupta, 2016) among others. We consider the following SDP for similarity based clustering. Let $S \in \mathbb{R}^{n \times n}$ be a symmetric similarity matrix among $n$ items, and $Z \in \{0, 1\}^{n \times k}$ be the cluster assignment matrix that we wish to estimate. For unknown number of clusters $k$, it is difficult to directly determine $Z$, and hence, we estimate the *normalised clustering matrix* $X \in \mathbb{R}^{n \times n}$ such that $X_{ij} = \frac{1}{|\mathcal{C}|}$ if $i, j$ co-occur in estimated cluster $\mathcal{C}$, and $X_{ij} = 0$ otherwise. Note that trace $(X) = k$. The following SDP was proposed and analysed by Yan et al. (2018) under the stochastic block model for graphs, and can also be applied in the more general context of data clustering (Chen and Yang, 2020). This SDP is agnostic to the number of clusters, but penalises large values of trace $(X)$ to restrict the number of estimated clusters:

$$
\max_{X} \; \text{trace}\,(SX) - \lambda\,\text{trace}\,(X)
$$
$$
\text{s.t. } X \geq 0, \quad X \succeq 0, \quad X\mathbf{1} = \mathbf{1}.
\tag{SDP-$\lambda$}
$$

Here, $\lambda$ is a tuning parameter and $\mathbf{1}$ denotes the vector of all ones. The constraints $X \geq 0$ and $X \succeq 0$ restricts the optimisation to non-negative, positive semi-definite matrices.

We first present a general theoretical result for SDP-$\lambda$. Assume that the data has an implicit partition into $k$ clusters $\mathcal{C}_1, \ldots, \mathcal{C}_k$ of sizes $n_1, \ldots, n_k$ and with cluster assignment matrix $Z$, and suppose that the similarity $S$ is close to an *ideal similarity matrix* $\widetilde{S}$ that has a $k \times k$ block structure $\widetilde{S} = Z\Sigma Z^T$. The matrix $\Sigma \in \mathbb{R}^{k \times k}$ is such that $\Sigma_{\ell\ell'}$ represents the ideal pairwise similarity between items from clusters $\mathcal{C}_\ell$ and $\mathcal{C}_{\ell'}$. Typically, under a random planted model, $\widetilde{S}$ is the same as $\mathbf{E}[S]$ up to possible differences in the diagonal terms. For $S = \widetilde{S}$ and certain values of $\lambda$, the unique optimal solution of SDP-$\lambda$ is a block diagonal matrix $X^* = ZN^{-1}Z^T$, where $N \in \mathbb{R}^{k \times k}$ is diagonal with entries $n_1, \ldots, n_k$ (see Appendix B). Thus, in the *ideal case*, solving the SDP provides the desired normalised clustering matrix from which one can recover the partition $\mathcal{C}_1, \ldots, \mathcal{C}_k$. The following result shows that $X^*$ is also the unique optimal solution of SDP-$\lambda$ if $S$ is sufficiently close to $\widetilde{S}$.

**Proposition 1 (Recovery of planted clusters using SDP-$\lambda$).** *Let $Z \in \{0, 1\}^{n \times k}$ be the assignments for a planted $k$-way clustering, $\widetilde{S} = Z\Sigma Z^T$, and $X^* = ZN^{-1}Z^T$ as defined above. Define*

$$
\Delta_1 = \min_{\ell \neq \ell'} \left( \frac{\Sigma_{\ell\ell} + \Sigma_{\ell'\ell'}}{2} - \Sigma_{\ell\ell'} \right), \quad \text{and} \quad \Delta_2 = \max_{i \in [n]} \max_{\ell \in [k]} \left| \frac{1}{|\mathcal{C}_\ell|} \sum_{j \in \mathcal{C}_\ell} \left( S_{ij} - \widetilde{S}_{ij} \right) \right|.
$$

*$X^*$ is the unique optimal solution of SDP-$\lambda$ for any choice of $\lambda$ in the interval*

$$\left\|S - \widetilde{S}\right\|_2 < \lambda < \min_\ell n_\ell \cdot \min\left\{\frac{\Delta_1}{2}, \, \Delta_1 - 6\Delta_2\right\}.$$

The proof of Proposition 1, given in Appendix B, is adapted from Yan et al. (2018) although uniqueness was not proved in this previous work. The term $\Delta_1$ quantifies the separation between the ideal within and inter-cluster similarities, and is similar in spirit to the weak assortativity criterion for stochastic block models (Yan et al., 2018). On the other hand, the matrix spectral norm $\|S - \widetilde{S}\|_2$ and the term $\Delta_2$ both quantify the deviation of the similarities $S$ from their ideal values $\widetilde{S}$. Note that the number of clusters can be computed as $k = \text{trace}(X)$ and cluster assignment $Z$ is obtained by clustering the rows of $X^*$ using $k$-means or spectral clustering for example. In the experiments (Section 5), we present a data-dependent approach to tune $\lambda$ and find $k$.

We conclude this section by noting that most of the previous analyses of SDP clustering either assume sub-Gaussian data (Yan and Sarkar, 2016) or consider similarity matrices with independence assumptions (Chen and Xu, 2014; Yan et al., 2018) that might not hold in general, and do not hold for our AddS-3 and AddS-4 similarities described in the next section. In contrast, the deterministic criteria stated in Proposition 1 make the result applicable in more general settings.

## 4  Similarities from passive comparisons

We present two new similarity functions computed from passive comparisons (AddS-3 and AddS-4) and guarantees for recovering planted clusters using SDP-$\lambda$ in conjunction with these similarities. Kleindessner and von Luxburg (2017) introduced pairwise similarities computed from triplets. A quadruplets variant was proposed by Ghoshdastidar et al. (2019). These similarities, detailed in Appendix A, are positive-definite kernels and have multiplicative forms. In contrast, we compute the similarity between items $i, j$ by simply adding binary responses to comparisons involving $i$ and $j$.

**Similarity from quadruplets.** We construct the additive similarity for quadruplets, referred to as AddS-4, in the following way. Recall the definition of $\mathcal{Q}$ in Equation (1) and for every $i \neq j$, define

$$S_{ij} = \sum_{r \neq s}\left(\mathbb{I}_{\{(i,j,r,s)\in\mathcal{Q}\}} - \mathbb{I}_{\{(r,s,i,j)\in\mathcal{Q}\}}\right), \tag{AddS-4}$$

where $\mathbb{I}_{\{.\}}$ is the indicator function. The intuition is that if $i, j$ are similar ($w_{ij}$ is large), then for every observed tuple $i, j, r, s$, $w_{ij} > w_{rs}$ is more likely to be observed. Thus, $(i, j, r, s)$ appears in $\mathcal{Q}$ more often than $(r, s, i, j)$, and $S_{ij}$ is a (possibly large) positive term. On the other hand, smaller $w_{ij}$ leads to a negative value of $S_{ij}$. Under the aforementioned planted model with clusters of size $n_1, \ldots, n_k$, one can verify that $S_{ij}$ indeed reveals the planted clusters in expectation since if $i, j$ belong to the same planted cluster, then $\mathbf{E}[S_{ij}] = p\epsilon\delta \sum_{\ell\in[k]} \frac{n_\ell(n - n_\ell)}{2}$, and $\mathbf{E}[S_{ij}] = -p\epsilon\delta \sum_{\ell\in[k]} \binom{n_\ell}{2}$ otherwise.

Thus, in expectation, the within cluster similarity exceeds the inter-cluster similarity by $p\epsilon\delta\binom{n}{2}$.

**Similarity from triplets.** The additive similarity based on passive triplets AddS-3 is given by

$$S_{ij} = \sum_{r \neq i,j}\left(\mathbb{I}_{\{(i,j,r)\in\mathcal{T}\}} - \mathbb{I}_{\{(i,r,j)\in\mathcal{T}\}}\right) + \left(\mathbb{I}_{\{(j,i,r)\in\mathcal{T}\}} - \mathbb{I}_{\{(j,r,i)\in\mathcal{T}\}}\right) \tag{AddS-3}$$

for every $i \neq j$. The AddS-3 similarity $S_{ij}$ aggregates all the comparisons that involve both $i$ and $j$, with either $i$ or $j$ as the reference item. Similar to the case of AddS-4, $S_{ij}$ tends to be positive when $w_{ij}$ is large, and negative for small $w_{ij}$. One can also verify that, under a planted model, the expected within cluster AddS-3 similarity exceeds the inter-cluster similarity by $p\epsilon\delta(n - 2)$.

A significant advantage of AddS-3 and AddS-4 over existing similarities is in terms of computational time for constructing $S$. Unlike existing kernels, both similarities can be computed from a single pass over $\mathcal{T}$ or $\mathcal{Q}$. In addition, the following result shows that the proposed similarities can exactly recover planted clusters using only a few (near optimal) number of passive comparisons.

**Theorem 1 (Cluster recovery using AddS-3 and AddS-4).** *Let $X^*$ denote the normalised cluster-ing matrix corresponding to the true partition, and $n_{\min}$ be the size of the smallest planted cluster.*

*Given the triplet or the quadruplet setting, there exist absolute constants $c_1, c_2, c_3, c_4 > 0$ such that, with probability at least $1 - \frac{1}{n}$, $X^*$ is the unique optimal solution of SDP-$\lambda$ if $\delta$ satisfies*

$$c_1 \frac{\sqrt{n \ln n}}{n_{\min}} < \delta \leq 1 \text{ , and one of the following two conditions hold:}$$

- *(triplet setting) $S$ is given by AddS-3, and the number of triplets $|\mathcal{T}|$ and the parameter $\lambda$ satisfy*

$$|\mathcal{T}| > c_2 \frac{n^3 (\ln n)^2}{\epsilon^2 \delta^2 n_{\min}^2} \quad and \quad c_3 \max\left\{ \sqrt{|\mathcal{T}| \frac{\ln n}{n}}, |\mathcal{T}| \epsilon \sqrt{\frac{\ln n}{n^3}}, (\ln n)^2 \right\} < \lambda < c_4 |\mathcal{T}| \frac{\epsilon \delta n_{\min}}{n^2} \text{ ;}$$

- *(quadruplet setting) $S$ is given by AddS-4, and the number of quadruplets $|\mathcal{Q}|$ and $\lambda$ satisfy*

$$|\mathcal{Q}| > c_2 \frac{n^3 (\ln n)^2}{\epsilon^2 \delta^2 n_{\min}^2} \quad and \quad c_3 \max\left\{ \sqrt{|\mathcal{Q}| \frac{\ln n}{n}}, |\mathcal{Q}| \epsilon \sqrt{\frac{\ln n}{n^3}}, (\ln n)^2 \right\} < \lambda < c_4 |\mathcal{Q}| \frac{\epsilon \delta n_{\min}}{n^2} \text{ .}$$

*The condition on $\delta$ and the number of comparisons ensure that the interval for $\lambda$ is non-empty.*

Theorem 1 is proved in Appendix C. This result shows that given a sufficient number of comparisons, one can exactly recover the planted clusters using SDP-$\lambda$ with an appropriate choice of $\lambda$. In particular, if there are $k$ planted clusters of similar sizes and $\delta$ satisfies the stated condition, then recovery of the planted clusters with zero error is possible with only $\Omega\left(\frac{k^2}{\epsilon^2 \delta^2} n (\ln n)^2\right)$ passively obtained triplets or quadruplets. In this particular context, we make a few important remarks about the sufficient conditions.

**Remark 1** (**Comparison with existing results**). For fixed $k$ and fixed $\epsilon, \delta \in (0, 1]$, Theorem 1 states that $\Omega\left(n(\ln n)^2\right)$ passive comparisons (triplets or quadruplets) suffice to exactly recover the clusters. This significantly improves over the result of Ghoshdastidar et al. (2019) stating that $\Omega\left(n^{3.5} \ln n\right)$ passive quadruplets are sufficient in a planted setting, and the fact that $\Omega\left(n^3\right)$ triplets are necessary in the worst case (Emamjomeh-Zadeh and Kempe, 2018).

**Remark 2** (**Dependence of the number of comparisons on the noise levels $\epsilon, \delta$**). When one can actively obtain comparisons, Emamjomeh-Zadeh and Kempe (2018) showed that it suffices to query $\Omega\left(n \ln\left(\frac{n}{\epsilon}\right)\right)$ triplets. Compared to the $\ln\left(\frac{1}{\epsilon}\right)$ dependence in the active setting, the sufficient number of passive comparisons in Theorem 1 has a stronger dependence of $\frac{1}{\epsilon^2}$ on the crowd noise level $\epsilon$. While we do not know whether this dependence is optimal, the stronger criterion is intuitive since, unlike the active setting, the passive setting does not provide repeated observations of the same comparisons that can easily nullify the crowd noise. The number of comparisons also depends as $\frac{1}{\delta^2}$ on the inherent noise level, which is similar to the conditions in Ghoshdastidar et al. (2019).

Theorem 1 states that exact recovery primarily depends on two sufficient conditions, one on $\delta$ and the other on the number of passive comparisons ($|\mathcal{T}|$ or $|\mathcal{Q}|$). The following two remarks show that both conditions are necessary, up to possible differences in logarithmic factors.

**Remark 3** (**Necessity of the condition on $\delta$**). The condition on $\delta$ imposes the condition of $n_{\min} = \Omega\left(\sqrt{n \ln n}\right)$. This requirement on $n_{\min}$ appears naturally in planted problems. Indeed, assuming that all $k$ clusters are of similar sizes, the above condition is equivalent to a requirement of $k = \mathcal{O}\left(\sqrt{\frac{n}{\ln n}}\right)$ and it is believed that polynomial time algorithms cannot recover $k \gg \sqrt{n}$ planted clusters (Chen and Xu, 2014, Conjecture 1).

**Remark 4** (**Near-optimal number of comparisons**). To cluster $n$ items, one needs to observe each example at least once. Hence, one trivially needs at least $\Omega(n)$ comparisons (active or passive). Similarly, existing works on actively obtained comparisons show that $\Omega(n \ln n)$ comparisons are sufficient for learning in supervised or unsupervised problems (Haghiri et al., 2017; Emamjomeh-Zadeh and Kempe, 2018; Ghoshdastidar et al., 2019). We observe that, in the setting of Remark 1, it suffices to have $\Omega\left(n(\ln n)^2\right)$ passive comparisons which matches the necessary conditions up to logarithmic factors. However, the sufficient condition on the number of comparisons becomes $\Omega\left(k^2 n(\ln n)^2\right)$ if $k$ grows with $n$ while $\epsilon$ and $\delta$ are fixed. It means that the worst case of $k = \mathcal{O}\left(\sqrt{\frac{n}{\ln n}}\right)$, stated in Remark 3, can only be tackled using at least $\Omega\left(n^2 \ln n\right)$ passive comparisons.

**Remark 5** (**No new information beyond $\Omega\left(n^2/\epsilon^2\right)$ comparisons**). Theorem 1 shows that for large $n$ and $\Omega\left(n^2/\epsilon^2\right)$ number of comparisons, the condition for exact recovery of the clusters is only

governed by the condition on $\delta$ as the interval for $\lambda$ is always non empty. It means that, beyond a quadratic number of comparisons, no new information is gained by observing more comparisons. This explains why significantly fewer passive comparisons suffice in practice than the known worst-case requirements of $\Omega\left(n^3\right)$ passive triplets or $\Omega\left(n^4\right)$ passive quadruplets.

We conclude our theoretical discussion with a remark about recovering planted clusters when the pairwise similarities $w_{ij}$ are observed. Our methods are near optimal even in this setting.

**Remark 6** (**Recovering planted clusters for non-parametric** $F_{in}, F_{out}$)**.** Theoretical studies in the classic setting of clustering with observed pairwise similarities $\{w_{ij} : i < j\}$ typically assume that the distributions $F_{in}$ and $F_{out}$ for the pairwise similarities are Bernoulli (in unweighted graphs), or take finitely many values (labelled graphs), or belong to exponential families (Chen and Xu, 2014; Aicher et al., 2015; Yun and Proutiere, 2016). Hence, the applicability of such results are restrictive. Recently, Xu et al. (2020) considered non-parametric distributions for $F_{in}, F_{out}$, and presented a near-optimal approach based on discretisation of the similarities into finitely many bins. Our work suggests an alternative approach: compute ordinal comparisons from the original similarities and use clustering on AddS-3 or AddS-4. Theorem 1 then guarantees, for any non-parametric and continuous $F_{in}$ and $F_{out}$, exact recovery of the planted clusters under a near-optimal condition on $\delta$.

## 5    Experiments

The goal of this section is three-fold: present a strategy to tune $\lambda$ in SDP-$\lambda$; empirically validate our theoretical findings; and demonstrate the performance of the proposed approaches on real datasets.

**Choosing $\lambda$ and estimating the number of clusters based on Theorem 1.** Given a similarity matrix $S$, the main difficulty involved in using SDP-$\lambda$ is tuning the parameter $\lambda$. Yan et al. (2018) proposed the algorithm SPUR to select the best $\lambda$ as $\lambda^* = \arg\max_{0 \leq \lambda \leq \lambda_{\max}} \frac{\sum_{i \leq k_\lambda} \sigma_i(X_\lambda)}{\text{trace}(X_\lambda)}$ where $X_\lambda$ is the solution of SDP-$\lambda$, $k_\lambda$ is the closest integer to $\text{trace}(X_\lambda)$ and an estimate of the number of clusters, $\sigma_i(X_\lambda)$ is the $i$-th largest eigenvalue of $X_\lambda$, and $\lambda_{\max}$ is a theoretically well-founded upper bound on $\lambda$. The maximum of the above objective is 1, achieved when $X_\lambda$ has the same structure as $X^*$ in Proposition 1. In our setting, Theorem 1 gives an upper bound on $\lambda$ that depends on $\epsilon$, $\delta$ and $n_{\min}$ which are not known in practice. Furthermore, it is computationally beneficial to use the theoretical lower bound for $\lambda$ instead of using $\lambda \geq 0$ as suggested in SPUR.

We propose to modify SPUR based on the fact that the estimated number of clusters $k$ monotonically decreases with $\lambda$ (details in Appendix D). Given Theorem 1, we choose $\lambda_{\min} = \sqrt{c(\ln n)/n}$ and $\lambda_{\max} = c/n$, where $c = |\mathcal{Q}|$ or $|\mathcal{T}|$. The trace of the SDP-$\lambda$ solution then gives two estimates of the number of clusters, $k_{\lambda_{\min}}$ and $k_{\lambda_{\max}}$, and we search over $k \in [k_{\lambda_{\max}}, k_{\lambda_{\min}}]$ instead of searching over $\lambda$—in practice, it helps to search over the values $\max\{2, k_{\lambda_{\max}}\} \leq k \leq k_{\lambda_{\min}} + 2$. We select $k$ that maximises the above SPUR objective, where $X$ is computed using a simpler SDP (Yan et al., 2018):

$$\max_X \langle S, X \rangle \qquad \text{s.t. } X \geq 0, \quad X \succeq 0, \quad X\mathbf{1} = \mathbf{1}, \quad \text{trace}(X) = k. \qquad \text{(SDP-}k\text{)}$$

The overall approach is summarized in Algorithm 1.

**Clustering with AddS-3 and AddS-4.**[3] For the proposed similarity matrices AddS-3 and AddS-4, the above strategy provides the optimal number of clusters $k$ and a corresponding solution $X_k$ of SDP-$k$. The partition is obtained by clustering the rows of $X_k$ using $k$-means. Alternative approaches, such as spectral clustering, lead to similar performances (see Appendix E).

**Evaluation function.** We use the Adjusted Rand Index (ARI) (Hubert and Arabie, 1985) between the ground truth and the predictions. The ARI takes values in $[-1, 1]$ and measures the agreement between two partitions: 1 implies identical partitions, whereas 0 implies that the predicted clustering is random. In all the experiments, we report the mean and standard deviation over 10 repetitions.

**Simulated data with planted clusters.** We generate data using the planted model from Section 2 and verify that the learned clusters are similar to the planted ones. As default parameters we use $n = 1000$, $k = 4$, $\epsilon = 0.75$, $|\mathcal{T}| = |\mathcal{Q}| = n(\ln n)^4$ and $F_{in} = \mathcal{N}\left(\sqrt{2}\sigma\Phi^{-1}\left(\frac{1+\delta}{2}\right), \sigma^2\right)$, $F_{out} = \mathcal{N}\left(0, \sigma^2\right)$ with $\sigma = 0.1$ and $\delta = 0.5$. In each experiment, we investigate the sensitivity of our method by varying one of the parameters while keeping the others fixed. We use SPUR to estimate the number of clusters.

**Algorithm 1:** Comparison-based SPUR

**input** : The number of examples $n$ and the comparisons $\mathcal{T}$ or $\mathcal{Q}$.

**begin**

&emsp;Define $c = |\mathcal{T}|$ or $|\mathcal{Q}|$.
&emsp;Let $S$ be obtained with AddS-3 or AddS-4.

&emsp;Define $\lambda_{\min} = \sqrt{\frac{c(\ln c)}{n}}$ and $\lambda_{\max} = \frac{c}{n}$.
&emsp;$X_{\lambda_{\min}}, X_{\lambda_{max}} \leftarrow$ SDP-$\lambda_{\min}$, SDP-$\lambda_{\max}$ on $S$.
&emsp;$k_{\lambda_{\min}}, k_{\lambda_{\max}} \leftarrow \lfloor \text{trace}(X_{\lambda_{\min}}) \rceil, \lfloor \text{trace}(X_{\lambda_{\max}}) \rceil$.
&emsp;**for** $k = \max\{2, k_{\lambda_{\max}}\}$ *to* $k_{\lambda_{min}} + 2$ **do**
&emsp;&emsp;| Solve SDP-$k$ to obtain $X_k$.
&emsp;**end**

&emsp;Choose $\hat{k} = \underset{k}{\text{argmax}} \frac{\sum_{i \leq k} \sigma_i(X_k)}{\text{trace}(X_k)}$, where $\sigma_i(X_k)$ denotes the $i$-th largest eigenvalue of $X_k$.

**end**

**output :** Number of clusters $\hat{k}$, $X_{\hat{k}}$.

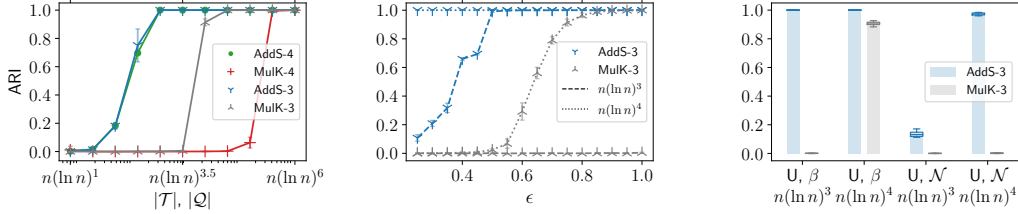

(a) Vary the number of comparisons &emsp; (b) Vary the external noise level, $\epsilon$ &emsp; (c) Vary the distributions $F_{in}$, $F_{out}$

Figure 1: ARI of various methods on the planted model (higher is better). We vary: (1a) the number of comparisons $|\mathcal{T}|$ and $|\mathcal{Q}|$; (1b) the crowd noise level $\epsilon$; (1c) the distributions $F_{in}$ and $F_{out}$.

As baselines, we use SDP-$k$ (using the number of clusters estimated by our approaches) followed by $k$-means with two comparison based multiplicative kernels: MulK-3 for triplets (Kleindessner and von Luxburg, 2017) and MulK-4 for quadruplets (Ghoshdastidar et al., 2019).

We present some significant results in Figure 1 and defer the others to Appendix E. In Figure 1a, we vary the number of sampled comparisons. Unsurprisingly, our approaches are able to exactly recover the planted clusters using as few as $n(\ln n)^3$ comparisons—extra $\ln n$ factor compared to Theorem 1 accounts for $\epsilon, \delta$ and constants. MulK-3 and MulK-4 respectively need $n(\ln n)^{4.5}$ and $n(\ln n)^{5.5}$ comparisons (both values exceed $n^2$ for $n = 1000$). In all our experiments, AddS-3 and AddS-4 have comparable performance while MulK-3 is significantly better than MulK-4. Thus we focus on triplets in the subsequent experiments for the sake of readability. In Figure 1b, we vary the external noise level $\epsilon$. Given $n(\ln n)^4$ comparisons, AddS-3 exactly recovers the planted clusters for $\epsilon$ as small as 0.25 (high crowd noise) while, given the same number of comparisons, MulK-3 only recovers the planted clusters for $\epsilon > 0.9$. Figure 1c shows that AddS-3 outperforms MulK-3 even when different distributions for $F_{\text{in}}$ and $F_{\text{out}}$ are considered (Uniform + Beta or Uniform + Normal; details in Appendix E). It also shows that the distributions affect the performances, which is not evident from Theorem 1, indicating the possibility of a refined analysis under distributional assumptions.

**MNIST clustering with comparisons.** We consider two datasets which are subsets of the MNIST test dataset (LeCun and Cortes, 2010) that originally contains 10000 examples roughly equally distributed among the ten digits: (i) a subset of 2163 examples containing all the 1 and 7 (*MNIST 1vs.7*), two digits that are visually very similar, and (ii) a randomly selected subset of 2000 examples drawn without replacement and covering all 10 classes (*MNIST 10*). In both cases, to generate the comparisons, we use the Gaussian similarity (See Section F.3 in the supplementary) on a 2-dimensional embedding of the entire MNIST test data constructed with t-SNE (van der Maaten, 2014) and normalized so that each example lies in $[-1, 1]^2$. We focus on the triplet setting and we randomly and uniformly draw, without replacement, between $n(\ln n)^2$ and $n(\ln n)^4$ comparisons to be observed by the different approaches. We also consider two additional baselines. First, we

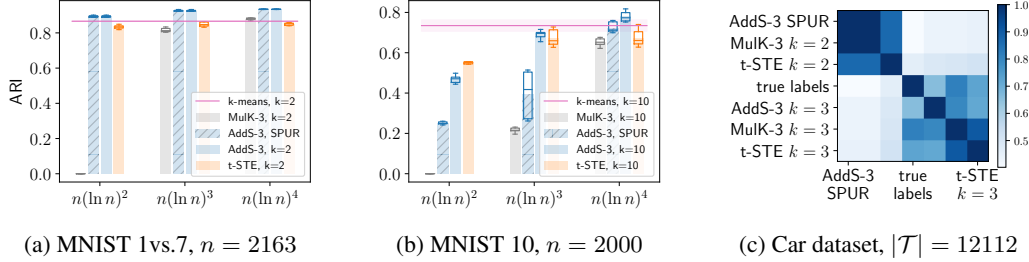

(a) MNIST 1vs.7, $n = 2163$      (b) MNIST 10, $n = 2000$      (c) Car dataset, $|\mathcal{T}| = 12112$

Figure 2: Experiments on real datasets. (2a)–(2b) ARI on MNIST; (2c) ARI similarity matrix comparing the clusters obtained by the different methods on car (darker means more agreement).

use t-STE (van der Maaten and Weinberger, 2012), an ordinal embedding approach, to embed the examples in 2 dimensions, and then cluster them using $k$-means on the embedded data. Second, we directly use $k$-means on the normalized data obtained with t-SNE. The latter is a baseline with access to Euclidean data instead of triplet comparisons.

For *MNIST 1vs.7* (Figure 2a), $|\mathcal{T}| = n(\ln n)^2$ is sufficient for AddS-3 to reach the performance of $k$-means and t-STE while MulK-3 requires $n(\ln n)^3$ triplets. Furthermore, note that AddS-3 with known number of clusters performs similarly to AddS-3 using SPUR, indicating that SPUR estimates the number of clusters correctly. If we consider *MNIST 10* (Figure 2b) and $|\mathcal{T}| = n(\ln n)^2$, AddS-3 with known $k$ outperforms AddS-3 using SPUR, suggesting that the number of comparisons here is not sufficient to estimate the number of clusters accurately. Moreover, AddS-3 with known $k$ outperforms MulK-3 while being close to the performance of t-STE. Finally for $n(\ln n)^4$ triplets, all ordinal methods converge to the baseline of $k$-means with access to original data. The ARI of AddS-3 SPUR improves when the number of comparisons increases due to better estimations of the number of clusters—estimated $k$ increases from 3 for $|\mathcal{T}| = n(\ln n)^2$ up to 9 for $|\mathcal{T}| = n(\ln n)^4$.

**Real comparison based data.** First, we consider the Food dataset (Wilber et al., 2014) that contains 100 examples and 190376 triplet comparisons. Unfortunately, there is no ground truth and, thus, quantitatively assessing the quality of the obtained partitions is difficult. Thus, in Appendix F, we simply compare the different methods with respect to one another and present the partition obtained by AddS-3 for visual inspection. Second, we consider the Car dataset (Kleindessner and von Luxburg, 2016). It contains 60 examples grouped into 3 classes (SUV, city cars, sport cars) with 4 outliers, and exhibits 12112 triplet comparisons. For this dataset, AddS-3 SPUR estimates $k = 2$ instead of the correct 3 clusters. Figure 2c considers all ordinal methods with $k = 2$ and $k = 3$, and shows the pairwise agreement (ARI) between different methods and also with the true labels. While MulK-3 with $k = 3$ agrees the most with the true labels, all the clustering methods agree well for $k = 2$ (top-left $3 \times 3$ block). Hence, the data may have another natural clustering with two clusters, suggesting possible discrepancies in how different people judge the similarities between cars (for instance, color or brand instead of the specified classes).

## 6 Conclusion

It is generally believed that a large number of passive comparisons is necessary in comparison based learning. Existing results on clustering require at least $\Omega\left(n^3\right)$ passive comparisons in the worst-case or under a planted framework. We show that, in fact, $\Omega\left(n(\ln n)^2\right)$ passive comparisons suffice for accurately recovering planted clusters. This number of comparisons is near-optimal, and almost matches the number of active comparisons typically needed for learning. Our theoretical findings are based on two simple approaches for constructing pairwise similarity matrices from passive comparisons (AddS-3 and AddS-4). The present analysis is in a restricted framework, where all within (or inter) cluster similarities are assumed to be identically distributed. Based on existing work on robustness of SDPs (Moitra et al., 2016), we believe that our theoretical result holds in a more general semi-random setting. Lastly, while we studied the merits of AddS-3 and AddS-4 in the context of clustering, they could be used for other problems such as semi-supervised learning, data embedding, or classification.

## Broader Impact

This work primarily has applications in the fields of psychophysics and crowdsourcing, and more generally, in learning from human responses. Such data and learning problems could be affected by implicit biases in human responses. However, this latter issue is beyond the scope of this work and, thus, was not formally analysed.

## Acknowledgments and Disclosure of Funding

The work of DG is partly supported by the Baden-Württemberg Stiftung through the BW Elitepro- gramm for postdocs. The work of MP has been supported by the ACADEMICS grant of the IDEXLYON, project of the Université de Lyon, PIA operated by ANR-16-IDEX-0005.

## Footnotes

[2]When we write that $\Omega\left(n(\ln n)^2\right)$ comparisons are sufficient, we express that any number of comparisons greater than $Cn(\ln n)^2$ with $C$ a constant is sufficient to solve the problem. In other words, it means that having exactly $Cn(\ln n)^2$ comparisons is sufficient but also that having more comparisons is not detrimental. This notation is used in statistics and information theory (Fletcher et al., 2009) and is equivalent to $\gtrsim$.

[3]We provide a Python implementation on `https://github.com/mperrot/AddS-Clustering`

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
