[Supplementary Material]

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

# A    Existing comparison based similarities / kernel functions

The literature on ordinal embedding from triplet comparisons is extensive (Jamieson and Nowak, 2011; Arias-Castro, 2017). In contrast, the idea of directly constructing similarity or kernel matrices from the comparisons, without embedding the data in an Euclidean space, is rather new. Such an approach is known to be significantly faster than embedding methods, and provides similar or sometimes better performances in certain learning tasks. To the best of our knowledge, there are only two works that learn kernel functions from comparisons (Kleindessner and von Luxburg, 2017; Ghoshdastidar et al., 2019), while the works of Jain et al. (2016) and Mason et al. (2017) estimate a Gram (or kernel) matrix from the triplets, which is then further used for data embedding. In this section, we describe the aforementioned approaches for constructing pairwise similarities from comparisons. Through this discussion, we illustrate the fundamental difference between the proposed additive similarities, AddS-3 and AddS-4, and the existing kernels that are of multiplicative nature (Kleindessner and von Luxburg, 2017; Ghoshdastidar et al., 2019).

Kernels from ordinal data were introduced by Kleindessner and von Luxburg (2017), who proposed two kernel functions (named $k_1$ and $k_2$) based on observed triplets. The kernels originated from the notion of Kendall's $\tau$ correlation between two rankings, and $k_1$ was empirically observed to perform slightly better. We mention this kernel function, which we refer to as a multiplicative triplet kernel (MulK-3). For any distinct $i, j \in [n]$, the MulK-3 similarity is computed as

$$S_{ij} = \frac{\sum\limits_{r<s} \left( \mathbb{I}_{\{(i,r,s)\in\mathcal{T}\}} - \mathbb{I}_{\{(i,s,r)\in\mathcal{T}\}} \right) \left( \mathbb{I}_{\{(j,r,s)\in\mathcal{T}\}} - \mathbb{I}_{\{(j,s,r)\in\mathcal{T}\}} \right)}{\sqrt{|\{(\ell, r, s) \in \mathcal{T} : \ell = i\}|}\sqrt{|\{(\ell, r, s) \in \mathcal{T} : \ell = j\}|}} \qquad \text{(MulK-3)}$$

where $\mathcal{T}$ is the set of observed triplets. Note that this kernel does not consider comparisons involving $w_{ij}$ but, instead, uses multiplicative terms indicating how $i$ and $j$ behave with respect to every pair $r, s$. For uniform sampling with rate $p \gg \frac{\ln n}{n^2}$, the denominators in MulK-3 are approximately $p\binom{n}{2}$ for every $i \neq j$. Hence, it suffices to focus only on the numerator. Ghoshdastidar et al. (2019) proposed a kernel similar to MulK-3 for the case of quadruplets, which is referred to as multiplicative quadruplet kernel (MulK-4). For $i \neq j$, it is given by

$$S_{ij} = \sum_{\ell\neq i,j} \sum_{r<s} \left( \mathbb{I}_{\{(i,\ell,r,s)\in\mathcal{Q}\}} - \mathbb{I}_{\{(r,s,i,\ell)\in\mathcal{Q}\}} \right) \left( \mathbb{I}_{\{(j,\ell,r,s)\in\mathcal{Q}\}} - \mathbb{I}_{\{(r,s,j,\ell)\in\mathcal{Q}\}} \right). \qquad \text{(MulK-4)}$$

Ghoshdastidar et al. (2019) studied MulK-4 in the context of hierarchical clustering, and showed that it requires $\mathcal{O}\left(n^{3.5} \ln n\right)$ passive quadruplet comparisons to exactly recover a planted hierarchical structure in the data. Combining their concentration results with Proposition 1 shows that the same number of passive quadruplets suffices to recover the planted clusters considered in this work. Note that both MulK-3 and MulK-4 kernel functions have a multiplicative nature since each entry is an aggregate of products. This is essential for their positive semi-definite property. In contrast, the proposed AddS-3 and AddS-4 similarities simply aggregate comparisons involving the pairwise similarity $w_{ij}$, and hence, are not positive semi-definite kernels.

We also mention the work on fast ordinal triplet embedding (FORTE) (Mason et al., 2017), which learns a metric from the given triplet comparisons. One can easily adapt the formulation to that of

learning a kernel matrix $K \in \mathbb{R}^{n \times n}$ from triplets. Consider the squared distance in the corresponding reproducing kernel Hilbert space (RKHS), $d_K^2(i,j) = K_{ii} - 2K_{ij} + K_{jj}$. Assuming that the triplets adhere to the distance relation in the RKHS, it is easy to see that when a comparison of $t = \{i, r, s\}$ with $i$ as pivot is available, then

$$y_t := \mathbb{I}_{\{(i,r,s) \in \mathcal{T}\}} - \mathbb{I}_{\{(i,s,r) \in \mathcal{T}\}} = \text{sign}\left(d_K^2(i,r) - d_K^2(i,s)\right)$$
$$= \text{sign}\left(K_{rr} - 2K_{ir} - K_{ss} + 2K_{is}\right),$$

which is the sign of a linear map of $K$, which we can denote as $\text{sign}(\langle M_t, K \rangle)$ for some $M_t \in \mathbb{R}^{n \times n}$. One can learn the optimal kernel matrix, that satisfies most triplet comparisons, by minimising the empirical loss $\frac{1}{|\mathcal{T}|} \sum_{t \in \mathcal{T}} \ell(y_t \langle M_t, K \rangle)$ with positive definiteness constraints for $K$, where $\ell$ is a loss function (log loss is suggested by Jain et al. (2016)).

# B  Proof of Proposition 1

In this section, we first provide a proof of Proposition 1 which is split into two parts: the proof of optimality of $X^*$, and the proof of uniqueness of the optimal solution. In addition, we provide a derivation for the claim that $X^*$ is the unique optimal solution for SDP-$\lambda$ when $S = \widetilde{S}$ and $0 < \lambda < n_{\min}\Delta_1$. The derivation, given at the end of the section, follows from simplifying some of the computations in the proof of Proposition 1.

## B.1  Optimality of $X^*$ when $S$ is close to $\widetilde{S}$

The proof is adapted from Yan et al. (2018). We first state the Karush-Kahn-Tucker (KKT) conditions for SDP-$\lambda$. Let $\Gamma, \Lambda \in \mathbb{R}^{n \times n}$ be the Lagrange parameters for the non-negativity constraint $(X \geq 0)$ and the positive semi-definiteness constraint $(X \succeq 0)$, respectively. Let $\alpha \in \mathbb{R}^n$ be the Lagrange parameter for the row sum constraints. The tuple $(X, \Lambda, \Gamma, \alpha)$ is a primal-dual optimal solution for SDP-$\lambda$ if and only if it satisfies the following KKT conditions:

| | |
|---|---|
| Stationarity : | $S - \lambda I + \Lambda + \Gamma - \mathbf{1}\alpha^T - \alpha\mathbf{1}^T = 0$ |
| Primal feasibility : | $X \geq 0 \quad ; \quad X \succeq 0 \quad ; \quad X\mathbf{1} = \mathbf{1}$ |
| Dual feasibility : | $\Lambda \succeq 0 \quad ; \quad \Gamma \geq 0$ |
| Complementary slackness : | $\langle \Lambda, X \rangle = 0 \quad ; \quad \Gamma_{ij}X_{ij} = 0 \ \forall \ i, j$ |

where we use $\langle A, B \rangle$ to denote trace $(AB)$ for symmetric matrices $A, B$. The above derivation is straightforward. The term $\mathbf{1}\alpha^T + \alpha\mathbf{1}^T$ in the stationarity condition arises due to the symmetry of $X$, that is, since row-sum and column-sum are identical. We construct a primal-dual witness to show that $X^*$ is the optimal solution of SDP-$\lambda$ under the stated conditions on $\lambda$. We use the following notations. For any vector $u \in \mathbb{R}^n$ and $\mathcal{C} \subset [n]$, we let $u_\mathcal{C} \in \mathbb{R}^{|\mathcal{C}|}$ be the projection of $u$ on the indices contained in $\mathcal{C}$. Similarly, for a matrix $A \in \mathbb{R}^{n \times n}$, $A_{\mathcal{C}\mathcal{C}'}$ is the sub-matrix corresponding to row indices in $\mathcal{C}$ and column indices in $\mathcal{C}'$. We also define $\mathbf{1}_m$ and $I_m$ the constant vector of ones and the identity matrix of size $m$, respectively. We use $\mathcal{C}_1, \dots, \mathcal{C}_k$ to denote the planted clusters of size $n_1, \dots, n_k$. We consider the following primal-dual construction that is similar to Yan et al. (2018), where $X = X^*$. For every $j, \ell \in \{1, \dots, k\}$ and $\ell \neq j$, we define

$$\alpha : \qquad \alpha_{\mathcal{C}_j} = \frac{1}{n_j} S_{\mathcal{C}_j \mathcal{C}_j} \mathbf{1}_{n_j} - \left(\frac{\lambda}{2n_j} + \frac{1}{2n_j^2} \mathbf{1}_{n_j}^T S_{\mathcal{C}_j \mathcal{C}_j} \mathbf{1}_{n_j}\right) \mathbf{1}_{n_j} \qquad (4)$$

$$\Lambda : \quad \begin{cases} \Lambda_{\mathcal{C}_j \mathcal{C}_j} = -S_{\mathcal{C}_j \mathcal{C}_j} + \alpha_{\mathcal{C}_j} \mathbf{1}_{n_j}^T + \mathbf{1}_{n_j} \alpha_{\mathcal{C}_j}^T + \lambda I_{n_j} \\ \Lambda_{\mathcal{C}_j \mathcal{C}_\ell} = -\left(I_{n_j} - \frac{1}{n_j} \mathbf{1}_{n_j} \mathbf{1}_{n_j}^T\right) S_{\mathcal{C}_j \mathcal{C}_\ell} \left(I_{n_\ell} - \frac{1}{n_\ell} \mathbf{1}_{n_\ell} \mathbf{1}_{n_\ell}^T\right) \end{cases} \qquad (5)$$

$$\Gamma : \quad \begin{cases} \Gamma_{\mathcal{C}_j \mathcal{C}_j} = 0 \\ \Gamma_{\mathcal{C}_j \mathcal{C}_\ell} = -S_{\mathcal{C}_j \mathcal{C}_\ell} - \Lambda_{\mathcal{C}_j \mathcal{C}_\ell} + \alpha_{\mathcal{C}_j} \mathbf{1}_{n_\ell}^T + \mathbf{1}_{n_j} \alpha_{\mathcal{C}_\ell}^T . \end{cases} \qquad (6)$$

The proof of Proposition 1 is based on verifying the KKT conditions for the above $\Lambda, \Gamma, \alpha$ and $X = X^*$. To this end, note that $X_{\mathcal{C}_j \mathcal{C}_j}^* = \frac{1}{n_j} \mathbf{1}_{n_j} \mathbf{1}_{n_j}^T$ and $X_{\mathcal{C}_j \mathcal{C}_\ell}^* = 0$ for $\ell \neq j$. Primal feasibility

is obviously satisfied by $X^*$, and it is easy to see that the choice of $\Lambda_{\mathcal{C}_j \mathcal{C}_j}$ and $\Gamma_{\mathcal{C}_j \mathcal{C}_\ell}$ ensures that stationarity holds. Hence, we only need to verify dual feasibility and complementary slackness.

The complementary slackness condition for $\Gamma$ holds since $\Gamma_{\mathcal{C}_j \mathcal{C}_j} = 0$ and $X^*_{\mathcal{C}_j \mathcal{C}_\ell} = 0$ for $j \neq \ell$. To verify $\langle \Lambda, X^* \rangle = 0$, observe that

$$\langle \Lambda, X^* \rangle = \sum_{j,\ell} \langle \Lambda_{\mathcal{C}_j \mathcal{C}_\ell}, X^*_{\mathcal{C}_j \mathcal{C}_\ell} \rangle = \sum_j \langle \Lambda_{\mathcal{C}_j \mathcal{C}_j}, X^*_{\mathcal{C}_j \mathcal{C}_j} \rangle = \sum_j \frac{1}{n_j} \mathbf{1}_{n_j}^T \Lambda_{\mathcal{C}_j \mathcal{C}_j} \mathbf{1}_{n_j}$$

$$= \sum_j -\frac{1}{n_j} \mathbf{1}_{n_j}^T S_{\mathcal{C}_j \mathcal{C}_j} \mathbf{1}_{n_j} + 2 \mathbf{1}_{n_j}^T \alpha_{\mathcal{C}_j} + \lambda,$$

where the last step follows by substituting $\Lambda_{\mathcal{C}_j \mathcal{C}_j}$ from (5) and noting that $\mathbf{1}_{n_j}^T \mathbf{1}_{n_j} = n_j$. Substituting the value of $\alpha_{\mathcal{C}_j}$ above shows that each term in the sum is zero, and hence, $\langle \Lambda, X^* \rangle = 0$.

We now verify the dual feasibility and first prove that $\Gamma \geq 0$, in particular, $\Gamma_{\mathcal{C}_j \mathcal{C}_\ell} \geq 0$ for $j \neq \ell$. We substitute $\Lambda_{\mathcal{C}_j \mathcal{C}_\ell}$ and $\alpha_{\mathcal{C}_j}$ in (6) to obtain

$$\Gamma_{\mathcal{C}_j \mathcal{C}_\ell} = -S_{\mathcal{C}_j \mathcal{C}_\ell} + \left( I_{n_j} - \frac{1}{n_j} \mathbf{1}_{n_j} \mathbf{1}_{n_j}^T \right) S_{\mathcal{C}_j \mathcal{C}_\ell} \left( I_{n_\ell} - \frac{1}{n_\ell} \mathbf{1}_{n_\ell} \mathbf{1}_{n_\ell}^T \right)$$

$$+ \frac{1}{n_j} S_{\mathcal{C}_j \mathcal{C}_j} \mathbf{1}_{n_j} \mathbf{1}_{n_\ell}^T - \left( \frac{\lambda}{2 n_j} + \frac{1}{2 n_j^2} \mathbf{1}_{n_j}^T S_{\mathcal{C}_j \mathcal{C}_j} \mathbf{1}_{n_j} \right) \mathbf{1}_{n_j} \mathbf{1}_{n_\ell}^T$$

$$+ \frac{1}{n_\ell} \mathbf{1}_{n_j} \mathbf{1}_{n_\ell}^T S_{\mathcal{C}_\ell \mathcal{C}_\ell} - \left( \frac{\lambda}{2 n_\ell} + \frac{1}{2 n_\ell^2} \mathbf{1}_{n_\ell}^T S_{\mathcal{C}_\ell \mathcal{C}_\ell} \mathbf{1}_{n_\ell} \right) \mathbf{1}_{n_j} \mathbf{1}_{n_\ell}^T$$

$$= -\frac{1}{n_j} \mathbf{1}_{n_j} \mathbf{1}_{n_j}^T S_{\mathcal{C}_j \mathcal{C}_\ell} - \frac{1}{n_\ell} S_{\mathcal{C}_j \mathcal{C}_\ell} \mathbf{1}_{n_\ell} \mathbf{1}_{n_\ell}^T + \frac{1}{n_j} S_{\mathcal{C}_j \mathcal{C}_j} \mathbf{1}_{n_j} \mathbf{1}_{n_\ell}^T + \frac{1}{n_\ell} \mathbf{1}_{n_j} \mathbf{1}_{n_\ell}^T S_{\mathcal{C}_\ell \mathcal{C}_\ell}$$

$$+ \left( \frac{\mathbf{1}_{n_j}^T S_{\mathcal{C}_j \mathcal{C}_\ell} \mathbf{1}_{n_\ell}}{n_j n_\ell} - \frac{\lambda}{2 n_j} - \frac{\mathbf{1}_{n_j}^T S_{\mathcal{C}_j \mathcal{C}_j} \mathbf{1}_{n_j}}{2 n_j^2} - \frac{\lambda}{2 n_\ell} - \frac{\mathbf{1}_{n_\ell}^T S_{\mathcal{C}_\ell \mathcal{C}_\ell} \mathbf{1}_{n_\ell}}{2 n_\ell^2} \right) \mathbf{1}_{n_j} \mathbf{1}_{n_\ell}^T .$$

Consider $i \in \mathcal{C}_j$ and $r \in \mathcal{C}_\ell$. From above, we can compute $\Gamma_{ir}$ as

$$\Gamma_{ir} = -\frac{1}{n_j} \mathbf{1}_{n_j}^T S_{\mathcal{C}_j r} - \frac{1}{n_\ell} S_{i \mathcal{C}_\ell} \mathbf{1}_{n_\ell} + \frac{1}{n_j} S_{i \mathcal{C}_j} \mathbf{1}_{n_j} + \frac{1}{n_\ell} \mathbf{1}_{n_\ell}^T S_{\mathcal{C}_\ell r}$$

$$+ \frac{\mathbf{1}_{n_j}^T S_{\mathcal{C}_j \mathcal{C}_\ell} \mathbf{1}_{n_\ell}}{n_j n_\ell} - \frac{\mathbf{1}_{n_j}^T S_{\mathcal{C}_j \mathcal{C}_j} \mathbf{1}_{n_j}}{2 n_j^2} - \frac{\mathbf{1}_{n_\ell}^T S_{\mathcal{C}_\ell \mathcal{C}_\ell} \mathbf{1}_{n_\ell}}{2 n_\ell^2} - \frac{\lambda}{2 n_j} - \frac{\lambda}{2 n_\ell}$$

$$= -\frac{1}{n_j} \sum_{i' \in \mathcal{C}_j} S_{i' r} - \frac{1}{n_\ell} \sum_{r' \in \mathcal{C}_\ell} S_{i r'} + \frac{1}{n_j} \sum_{i' \in \mathcal{C}_j} S_{i i'} + \frac{1}{n_\ell} \sum_{r' \in \mathcal{C}_\ell} S_{r r'}$$

$$+ \frac{1}{n_j n_\ell} \sum_{i' \in \mathcal{C}_j, r' \in \mathcal{C}_\ell} S_{i' r'} - \frac{1}{2 n_j^2} \sum_{i, i' \in \mathcal{C}_j} S_{i i'} - \frac{1}{2 n_\ell^2} \sum_{r, r' \in \mathcal{C}_\ell} S_{r r'} - \frac{\lambda}{2 n_j} - \frac{\lambda}{2 n_\ell} .$$

Our goal is to derive a lower bound for $\Gamma_{ir}$ and show that, for suitable values of $\lambda$, $\Gamma_{ir} \geq 0$ for all $i \in \mathcal{C}_j, r \in \mathcal{C}_\ell$. We bound each of the terms from below. For the last two terms involving $\lambda$, we note that both terms are at least $-\frac{\lambda}{2 n_{\min}}$, where $n_{\min} = \min_\ell n_\ell$. For each of the other terms, we rewrite the summations in terms of the ideal similarity matrix $\widetilde{S}$ and bound the deviation in terms of

$\Delta_2 = \max\limits_{i \in [n]} \max\limits_{\ell \in [k]} \left| \frac{1}{n_\ell} \sum\limits_{r \in \mathcal{C}_\ell} \left( S_{ir} - \widetilde{S}_{ir} \right) \right|$. For the first term, we have

$$-\frac{1}{n_j} \sum_{i' \in \mathcal{C}_j} S_{i' r} = -\frac{1}{n_j} \sum_{i' \in \mathcal{C}_j} \widetilde{S}_{i' r} - \frac{1}{n_j} \sum_{i' \in \mathcal{C}_j} \left( S_{i' r} - \widetilde{S}_{i' r} \right)$$

$$= -\Sigma_{j\ell} - \frac{1}{n_j} \sum_{i' \in \mathcal{C}_j} \left( S_{i' r} - \widetilde{S}_{i' r} \right)$$

$$\geq -\Sigma_{j\ell} - \Delta_2.$$

For the second inequality, we use the structure of $\widetilde{S}$ to note that $\widetilde{S}_{ir} = \Sigma_{j\ell}$ for every $i \in \mathcal{C}_i, r \in \mathcal{C}_\ell$, and finally the deviation term is bounded by $\Delta_2$. Similarly, one can bound the second, third and fourth terms from below by $(-\Sigma_{j\ell} - \Delta_2)$, $(\Sigma_{jj} - \Delta_2)$ and $(\Sigma_{\ell\ell} - \Delta_2)$, respectively. For the fifth term, we write

$$\frac{1}{n_j n_\ell} \sum_{i' \in \mathcal{C}_j, r' \in \mathcal{C}_\ell} S_{i'r'} = \frac{1}{n_j n_\ell} \sum_{i' \in \mathcal{C}_j, r' \in \mathcal{C}_\ell} \widetilde{S}_{i'r'} + \frac{1}{n_j n_\ell} \sum_{i' \in \mathcal{C}_j, r' \in \mathcal{C}_\ell} \left( S_{i'r'} - \widetilde{S}_{i'r'} \right)$$

$$= \Sigma_{j\ell} + \frac{1}{n_j} \sum_{i' \in \mathcal{C}_j} \left( \frac{1}{n_\ell} \sum_{r' \in \mathcal{C}_\ell} \left( S_{i'r} - \widetilde{S}_{i'r} \right) \right)$$

$$\geq \Sigma_{j\ell} - \Delta_2,$$

since each term in the outer summation is at least $-\Delta_2$. Similarly, one can bound the sixth and seventh terms from below by $\frac{1}{2}(\Sigma_{jj} - \Delta_2)$ and $\frac{1}{2}(\Sigma_{\ell\ell} - \Delta_2)$, respectively. Combining the above lower bounds, we have

$$\Gamma_{ir} \geq \frac{1}{2}\Sigma_{jj} + \frac{1}{2}\Sigma_{\ell\ell} - \Sigma_{j\ell} - 6\Delta_2 - \frac{\lambda}{n_{\min}} \geq (\Delta_1 - 6\Delta_2) - \frac{\lambda}{n_{\min}},$$

where we recall that $\Delta_1 = \min_{\ell \neq \ell'} \left( \frac{\Sigma_{\ell\ell} + \Sigma_{\ell'\ell'}}{2} - \Sigma_{\ell\ell'} \right)$. Hence, for $\lambda \leq n_{\min}(\Delta_1 - 6\Delta_2)$, as stated in Proposition 1, $\Gamma_{ir} \geq 0$, and more generally, $\Gamma$ is non-negative.

We finally derive the positive semi-definiteness of $\Lambda$. Define the vectors $u_1, \ldots, u_k \in \mathbb{R}^n$ such that $(u_\ell)_i = 1$ if $i \in \mathcal{C}_\ell$ and 0 otherwise. We first claim that $u_1, \ldots, u_k$ lie in the null space of $\Lambda$. To verify this, we compute the $\mathcal{C}_j$-th block of $\Lambda u_\ell$. For $j \neq \ell$,

$$(\Lambda u_\ell)_{\mathcal{C}_j} = \Lambda_{\mathcal{C}_j \mathcal{C}_\ell} \mathbf{1}_{n_\ell} = -\left( I_{n_j} - \frac{1}{n_j} \mathbf{1}_{n_j} \mathbf{1}_{n_j}^T \right) S_{\mathcal{C}_j \mathcal{C}_\ell} \left( I_{n_\ell} - \frac{1}{n_\ell} \mathbf{1}_{n_\ell} \mathbf{1}_{n_\ell}^T \right) \mathbf{1}_{n_\ell} = 0,$$

whereas for $j = \ell$, we have from (4) and (5),

$$(\Lambda u_\ell)_{\mathcal{C}_\ell} = \Lambda_{\mathcal{C}_\ell \mathcal{C}_\ell} \mathbf{1}_{n_\ell}$$

$$= -S_{\mathcal{C}_\ell \mathcal{C}_\ell} \mathbf{1}_{n_\ell} + n_\ell \alpha_{\mathcal{C}_\ell} + \mathbf{1}_{n_\ell} \alpha_{\mathcal{C}_\ell}^T \mathbf{1}_{n_\ell} + \lambda \mathbf{1}_{n_\ell}$$

$$= -S_{\mathcal{C}_\ell \mathcal{C}_\ell} \mathbf{1}_{n_\ell} + S_{\mathcal{C}_\ell \mathcal{C}_\ell} \mathbf{1}_{n_\ell} - 2\left( \frac{\lambda}{2} + \frac{\mathbf{1}_{n_\ell}^T S_{\mathcal{C}_\ell \mathcal{C}_\ell} \mathbf{1}_{n_\ell}}{2n_\ell} \right) \mathbf{1}_{n_\ell} + \mathbf{1}_{n_\ell} \frac{\mathbf{1}_{n_\ell}^T S_{\mathcal{C}_\ell \mathcal{C}_\ell} \mathbf{1}_{n_\ell}}{n_\ell} + \lambda \mathbf{1}_{n_\ell}$$

$$= 0.$$

Thus $\Lambda u_\ell = 0$ for $\ell = 1, \ldots, k$, and to prove that $\Lambda \succeq 0$, it suffices to show that $u^T \Lambda u \geq 0$ for all $u \in \mathbb{R}^n$ that are orthogonal to $u_1, \ldots, u_k$. In other words, we consider only $u$ such that $u_{\mathcal{C}_\ell}^T \mathbf{1}_{n_\ell} = 0$ for every $\ell$. For such a vector $u$, we have

$$u^T \Lambda u = \sum_{j,\ell=1}^{k} u_{\mathcal{C}_j}^T \Lambda_{\mathcal{C}_j \mathcal{C}_\ell} u_{\mathcal{C}_\ell} = \sum_{j=1}^{k} u_{\mathcal{C}_j}^T \Lambda_{\mathcal{C}_j \mathcal{C}_j} u_{\mathcal{C}_j} + \sum_{j \neq \ell} u_{\mathcal{C}_j}^T \Lambda_{\mathcal{C}_j \mathcal{C}_\ell} u_{\mathcal{C}_\ell}$$

$$= \sum_{j=1}^{k} u_{\mathcal{C}_j}^T (-S_{\mathcal{C}_j \mathcal{C}_j} + \lambda I_{n_j}) u_{\mathcal{C}_j} - \sum_{j \neq \ell} u_{\mathcal{C}_j}^T S_{\mathcal{C}_j \mathcal{C}_\ell} u_{\mathcal{C}_\ell}$$

$$= \sum_{j=1}^{k} \lambda u_{\mathcal{C}_j}^T u_{\mathcal{C}_j} - \sum_{j,\ell} u_{\mathcal{C}_j}^T S_{\mathcal{C}_j \mathcal{C}_\ell} u_{\mathcal{C}_\ell}$$

$$= \lambda \|u\|^2 - u^T S u,$$

where $\|u\|$ is the Euclidean norm. The third equality follows from (5) and $u_{\mathcal{C}_\ell}^T \mathbf{1}_{n_\ell} = 0$ for every $\ell$. In addition to above, recall that $\widetilde{S} = Z\Sigma Z^T$, where $Z = [u_1 \ldots u_k]$. Hence, for $u$ orthogonal to $u_1, \ldots, u_k$, we have $u^T \widetilde{S} u = 0$, which, combined with above, gives

$$u^T \Lambda u = \lambda \|u\|^2 - u^T S u$$

$$
\begin{aligned}
&= \lambda \|u\|^2 - u^T \left( S - \widetilde{S} \right) u \\
&\geq \left( \lambda - \left\| S - \widetilde{S} \right\|_2 \right) \|u\|^2 \\
&> 0
\end{aligned}
$$

for all $u$ if $\lambda > \left\| S - \widetilde{S} \right\|_2$, which is the condition stated in Proposition 1. Thus, for the specified range of $\lambda$, the KKT conditions are satisfied and $X^*$ is the optimal solution for SDP-$\lambda$.

## B.2  Uniqueness of the optimal solution $X^*$

The uniqueness of the solution can be shown by proving that any other optimal solution $X'$ for SDP-$\lambda$ must satisfy $X' = X^*$. This is shown in two steps. First, we show that any optimal solution $X'$ must have the same block structure as $X^*$ and $X^* - X' \succeq 0$. We use this fact to show that the objective value for $X^*$ is strictly greater than that for any such $X'$.

Note that the previously constructed Lagrange parameters in (4)–(6) need not correspond to the optimal solution associated with $X'$. However, for the previously defined $\alpha, \Lambda, \Gamma$, we can still use the condition for stationarity to write

$$
\begin{aligned}
\langle \Lambda + \Gamma, X' \rangle &= \langle -S + \mathbf{1}\alpha^T + \alpha \mathbf{1}^T + \lambda I, X' \rangle \\
&= -\langle S, X' \rangle + \sum_{i,j=1}^{n} (\alpha_i + \alpha_j) X'_{ij} + \lambda \mathrm{trace}\,(X) \\
&= -\mathrm{trace}\,(SX') + \lambda \mathrm{trace}\,(X') + 2\sum_{i=1}^{n} \alpha_i
\end{aligned}
$$

where the simplification happens noting that $X'$ is primal feasible and hence $\sum_j X_{ij} = 1$. Due to optimality of $X'$ and $X^*$, we have $\mathrm{trace}\,(SX') - \lambda\mathrm{trace}\,(X') = \mathrm{trace}\,(SX^*) - \lambda\mathrm{trace}\,(X^*)$, and so,

$$
\begin{aligned}
\langle \Lambda + \Gamma, X' \rangle &= -\mathrm{trace}\,(SX^*) + \lambda \mathrm{trace}\,(X^*) + 2\mathbf{1}^T \alpha \\
&= \lambda + \sum_{j=1}^{k} \left( -\frac{\mathbf{1}_{n_j}^T S_{\mathcal{C}_j \mathcal{C}_j} \mathbf{1}_{n_j}}{n_j} + 2\mathbf{1}_{n_j}^T \alpha_{\mathcal{C}_j} \right) = 0,
\end{aligned}
$$

where the final step follows by substituting $\alpha_{\mathcal{C}_j}$ from (4). From above, we argue that both $\langle \Lambda, X' \rangle$ and $\langle \Gamma, X' \rangle$ are zero. To verify this, note that $\Gamma$ and $X'$ are both non-negative and hence, $\langle \Gamma, X' \rangle \geq 0$. On the other hand, from the definition of Frobenius (or Hilbert-Schmidt) norm, we have $\langle \Lambda, X' \rangle = \left\| \Lambda^{1/2} X'^{1/2} \right\|_F^2 \geq 0$, where the matrices square roots exist since $\Lambda, X'$ are both positive semi-definite. Since both inner products, $\langle \Lambda, X' \rangle$ and $\langle \Gamma, X' \rangle$, are non-negative and yet their sum is zero, we can conclude that each of them equals zero.

Note that $\langle \Lambda, X' \rangle = \left\| \Lambda^{1/2} X'^{1/2} \right\|_F^2 = 0$ implies $\Lambda X' = 0$, or the range space of $X'$ lies in the null space of $\Lambda$. Recall, from the proof of positive semi-definiteness of $\Lambda$, that, for $\lambda > \|S - \widetilde{S}\|_2$, the null space of $\Lambda$ is exactly spanned by $Z = [u_1 \ldots u_k]$. Thus, the range space of $X'$ is spanned by the columns of $Z$, or in other words $X' = ZAZ^T$ for some $A \in \mathbb{R}^{k \times k}$ that is symmetric, non-negative, and positive semi-definite (to ensure that $X'$ is primal feasible), and $\sum_j A_{ij} n_j = 1$ (to satisfy the row sum constraint). Recall that $X^* = ZN^{-1}Z^T$ where $N = \mathrm{diag}(n_1, \ldots, n_k)$. Thus, $X'$ has the same block structure as $X^*$. However, this result does not imply that we can recover $k$ planted clusters from $X'$ since it is possible that $A$ has less than $k$ distinct rows.

We now argue that $X^* - X'$ must be positive semi-definite, a property that we use later. To see this, note that

$$
X^* - X' = ZN^{-1/2} \left( I_k - N^{1/2}AN^{1/2} \right) N^{-1/2}Z^T,
$$

where $ZN^{-1/2}$ is a matrix with orthonormal columns. Hence, to prove that $X^* - X' \succeq 0$, it suffices to show that $I_k - N^{1/2}AN^{1/2} \succeq 0$ or, equivalently, that the largest eigenvalue of $N^{1/2}AN^{1/2}$ is smaller than 1. This can verified as

$$
\left\| N^{1/2}AN^{1/2} \right\|_2 = \max_{u\,:\,\|u\|=1} u^T N^{1/2}AN^{1/2} u = \max_{u\,:\,\|u\|=1} \sum_{i,j=1}^{k} A_{ij}\sqrt{n_i n_j}\, u_i u_j.
$$

From the AM-GM inequality, we have $\sqrt{n_i n_j} u_i u_j \leq \frac{1}{2} \left( n_i u_j^2 + n_j u_i^2 \right)$. Hence,

$$\left\| N^{1/2} A N^{1/2} \right\|_2 \leq \max_{u\,:\,\|u\|=1} \frac{1}{2} \sum_{i,j=1}^{k} A_{ij} \left( n_i u_j^2 + n_j u_i^2 \right) = \sum_{i=1}^{k} u_i^2 = 1,$$

where we use the fact that $\sum_j A_{ij} n_j = \sum_i A_{ij} n_i = 1$. From this discussion, we have $X^* - X' \succeq 0$. We now claim that

$$\left| \text{trace} \left( (S - \widetilde{S})(X^* - X) \right) \right| \leq \left\| S - \widetilde{S} \right\|_2 \text{trace}\left( X^* - X' \right), \qquad (7)$$

which follows from von Neumann's trace inequality and the fact that $X^* - X'$ is positive semi-definite.

We now prove that for any $X' = ZAZ^T \neq X^*$, with $A$ satisfying the above mentioned conditions, and for $\|S - \widetilde{S}\|_2 < \lambda < \frac{1}{2}\Delta_1 n_{\min}$,

$$\text{trace}\left( SX^* \right) - \lambda \text{trace}\left( X^* \right) > \text{trace}\left( SX' \right) - \lambda \text{trace}\left( X' \right), \qquad (8)$$

which shows that $X^*$ is the unique optimal solution. We compute

$$\text{trace}\left( SX^* \right) - \lambda \text{trace}\left( X^* \right) - \text{trace}\left( SX' \right) + \lambda \text{trace}\left( X' \right)$$
$$= \text{trace}\left( S(X^* - X') \right) - \lambda \text{trace}\left( X^* - X' \right)$$
$$= \text{trace}\left( \widetilde{S}(X^* - X') \right) + \text{trace}\left( (S - \widetilde{S})(X^* - X') \right) - \lambda \text{trace}\left( X^* - X' \right)$$
$$> \text{trace}\left( \widetilde{S}(X^* - X') \right) - \left\| S - \widetilde{S} \right\|_2 \text{trace}\left( X^* - X' \right) - \lambda \text{trace}\left( X^* - X' \right)$$
$$> \text{trace}\left( \widetilde{S}(X^* - X') \right) - n_{\min}\Delta_1 \text{trace}\left( X^* - X' \right).$$

In the last step, we use $\left\| S - \widetilde{S} \right\|_2 + \lambda < 2\lambda < n_{\min}\Delta_1$. We later prove that

$$\text{trace}\left( \widetilde{S}(X^* - X') \right) \geq \sum_{\ell=1}^{k} n_\ell (1 - A_{\ell\ell} n_\ell)\Delta_1 \geq n_{\min}\Delta_1 \text{trace}\left( X^* - X' \right). \qquad (9)$$

Using (9) in the previous derivation proves (8) or the fact that $X^*$ is the unique optimal solution, provided that $\text{trace}\left( X^* - X' \right) > 0$ for all $X' \neq X^*$. Hence, we need to verify the strict positivity of the trace. Assume that $\text{trace}\left( X^* - X' \right) = 0$. Due to the row sum constraint for $X'$, we have $\sum_j A_{\ell j} n_j = 1$, which implies $A_{\ell\ell} n_\ell \leq 1$. On the other hand $\text{trace}\left( X' \right) = \text{trace}\left( X^* \right) = k$ holds if $\sum_\ell A_{\ell\ell} n_\ell = k$, which is only possible if $A_{\ell\ell} n_\ell = 1$ for every $\ell$, and hence $A_{\ell j} = 0$ for $j \neq \ell$. Thus, $\text{trace}\left( X^* - X' \right) = 0$ if and only if $X' = X^*$. For every $X' \neq X^*$, we have $\text{trace}\left( X^* - X' \right) = \sum_\ell (1 - A_{\ell\ell} n_\ell) > 0$. We conclude the proof by proving (9). We compute

$$\text{trace}\left( \widetilde{S}(X^* - X') \right) = \text{trace}\left( Z\Sigma Z^T Z(N^{-1} - A)Z^T \right)$$
$$= \text{trace}\left( \Sigma Z^T Z(N^{-1} - A)Z^T Z \right)$$
$$= \text{trace}\left( \Sigma N(N^{-1} - A)N \right)$$
$$= \sum_{\ell=1}^{k} \Sigma_{\ell\ell} n_\ell (1 - A_{\ell\ell} n_\ell) - \sum_{\ell=1}^{k} \sum_{j\neq\ell} A_{\ell j} n_j n_\ell \Sigma_{\ell j},$$

where the third equality follows from the fact $Z^T Z = N$. Recall from the definition of $\Delta_1$ that $\Sigma_{\ell j} \leq \frac{1}{2}(\Sigma_{jj} + \Sigma_{\ell\ell}) - \Delta_1$. Using this, we can write

$$\text{trace}\left( \widetilde{S}(X^* - X') \right)$$
$$\geq \sum_{\ell=1}^{k} \Sigma_{\ell\ell} n_\ell (1 - A_{\ell\ell} n_\ell) + \sum_{\ell=1}^{k} \sum_{j\neq\ell} A_{\ell j} n_j n_\ell \Delta_1 - \frac{1}{2} \sum_{\ell=1}^{k} \sum_{j\neq\ell} A_{\ell j} n_j n_\ell (\Sigma_{\ell\ell} + \Sigma_{jj})$$
$$= \sum_{\ell=1}^{k} \Sigma_{\ell\ell} n_\ell (1 - A_{\ell\ell} n_\ell) + \sum_{\ell=1}^{k} \sum_{j\neq\ell} A_{\ell j} n_j n_\ell \Delta_1 - \sum_{\ell=1}^{k} \sum_{j\neq\ell} A_{\ell j} n_j n_\ell \Sigma_{\ell\ell}$$

$$= \sum_{\ell=1}^{k} \Sigma_{\ell\ell} n_\ell (1 - A_{\ell\ell} n_\ell) + \sum_{\ell=1}^{k} n_\ell \Delta_1 (1 - A_{\ell\ell} n_\ell) - \sum_{\ell=1}^{k} n_\ell \Sigma_{\ell\ell} (1 - A_{\ell\ell} n_\ell).$$

In the first equality, we exploit the symmetry of the third summation, while the second equality uses the row sum constraint to write $\sum_{j \neq \ell} A_{\ell j} n_j = 1 - A_{\ell\ell} n_\ell$. Cancelling first and third terms, we get

$$\text{trace}\left( \widetilde{S}(X^* - X') \right) \geq \Delta_1 \sum_{\ell=1}^{k} n_\ell (1 - A_{\ell\ell} n_\ell)$$

$$\geq \Delta_1 n_{\min} \sum_{\ell=1}^{k} (1 - A_{\ell\ell} n_\ell) = n_{\min} \Delta_1 \text{trace}\left( X^* - X' \right),$$

which proves (9), and completes the proof.

### B.3 Unique optimality of $X^*$ when $S = \widetilde{S}$

We now prove that $X^*$ is the unique optimal solution when $S = \widetilde{S} = Z\Sigma Z^T$ and $0 < \lambda < n_{\min} \Delta_1$. This claim does not immediately follow from Proposition 1, but can be derived from the proof.

We first prove the optimality of $X^*$ in this case. Recall, from the proof of Proposition 1, that the claim hinges on showing that $\Gamma \geq 0$ and $\Lambda \succeq 0$. From the previous proof, it suffices to show that $\Gamma_{\mathcal{C}_j \mathcal{C}_\ell} \geq 0$ and $u^T \Lambda u \geq 0$ for any $u$ that is orthogonal to the columns of $Z$. To show that the latter holds, recall that

$$u^T \Lambda u = \lambda \|u\|^2 - u^T S u.$$

Since $S = \widetilde{S} = Z\Sigma Z^T$ and $Z^T u = 0$, we get $u^T \Lambda u = \lambda \|u\|^2 \geq 0$, which in turn shows that $\Lambda \succeq 0$ for all $\lambda > 0$. To verify the non-negativity of $\Gamma_{\mathcal{C}_j \mathcal{C}_\ell}$, we observe that, in this case, it can be computed as

$$\Gamma_{\mathcal{C}_j \mathcal{C}_\ell} = -\frac{1}{n_j} \mathbf{1}_{n_j} \mathbf{1}_{n_j}^T \widetilde{S}_{\mathcal{C}_j \mathcal{C}_\ell} - \frac{1}{n_\ell} \widetilde{S}_{\mathcal{C}_j \mathcal{C}_\ell} \mathbf{1}_{n_\ell} \mathbf{1}_{n_\ell}^T + \frac{1}{n_j} \widetilde{S}_{\mathcal{C}_j \mathcal{C}_j} \mathbf{1}_{n_j} \mathbf{1}_{n_\ell}^T + \frac{1}{n_\ell} \mathbf{1}_{n_j} \mathbf{1}_{n_\ell}^T \widetilde{S}_{\mathcal{C}_\ell \mathcal{C}_\ell}$$

$$+ \left( \frac{\mathbf{1}_{n_j}^T \widetilde{S}_{\mathcal{C}_j \mathcal{C}_\ell} \mathbf{1}_{n_\ell}}{n_j n_\ell} - \frac{\lambda}{2 n_j} - \frac{\mathbf{1}_{n_j}^T \widetilde{S}_{\mathcal{C}_j \mathcal{C}_j} \mathbf{1}_{n_j}}{2 n_j^2} - \frac{\lambda}{2 n_\ell} - \frac{\mathbf{1}_{n_\ell}^T \widetilde{S}_{\mathcal{C}_\ell \mathcal{C}_\ell} \mathbf{1}_{n_\ell}}{2 n_\ell^2} \right) \mathbf{1}_{n_j} \mathbf{1}_{n_\ell}^T$$

$$= \left( -2\Sigma_{j\ell} + \Sigma_{jj} + \Sigma_{\ell\ell} + \Sigma_{j\ell} - \frac{\lambda}{2} \left( \frac{1}{n_j} + \frac{1}{n_\ell} \right) - \frac{\Sigma_{jj} + \Sigma_{\ell\ell}}{2} \right) \mathbf{1}_{n_j} \mathbf{1}_{n_\ell}^T$$

$$\geq \left( \Delta_1 - \frac{\lambda}{n_{\min}} \right) \mathbf{1}_{n_j} \mathbf{1}_{n_\ell}^T$$

So for $\lambda \leq n_{\min} \Delta_1$, $\Gamma_{\mathcal{C}_j \mathcal{C}_\ell} \geq 0$, and hence $\Gamma$ is non-negative. Combining this with the previous proof of optimality, we derive that $X^*$ is an optimal solution in this case for $0 < \lambda \leq n_{\min} \Delta_1$.

The proof of uniqueness is similar to the more general case in Proposition 1. We use the previously derived claim that any optimal solution $X'$ must be of the form $X' = ZAZ^T$ for some $A \in \mathbb{R}^{k \times k}$, and $X^* - X \succeq 0$. We have also previously shown that $\text{trace}\left( \widetilde{S}(X^* - X') \right) \geq n_{\min} \Delta_1 \text{trace}\left( X^* - X' \right)$. Hence, we have

$$\text{trace}\left( \widetilde{S} X^* \right) - \lambda \text{trace}\left( X^* \right) - \left( \text{trace}\left( \widetilde{S} X' \right) - \lambda \text{trace}\left( X' \right) \right) \geq (n_{\min} \Delta_1 - \lambda) \text{trace}\left( X^* - X' \right),$$

which is strictly positive for $\lambda < n_{\min} \Delta_1$, and hence, $X^*$ is the unique optimal solution in this case.

## C  Proof of Theorem 1

We prove the result for triplets and quadruplets in separate sections. While the proof structure is the same in both cases, the computations are quite different. Before presenting the proofs, we list the key steps.

We first compute the expectation of the similarity matrix $S$ computed using AddS-3 or AddS-4, and derive appropriate ideal matrices $\widetilde{S}$ in each case. In our proofs, $\widetilde{S} = \mathbf{E}[S]$, except differences in the diagonal entries since $S_{ii} = 0$ for all $i$. From the block structure of $\widetilde{S}$, we can compute $\Delta_1$.

Subsequently, concentration inequalities are used to derive upper bounds on $\|S - \widetilde{S}\|_2$ and $\Delta_2$ in terms of the model parameters. In this context, note that though the pairwise similarities $\{w_{ij} \; : \; i < j\}$ are independent, the entries of the matrix $S$ are highly dependent since each $w_{ij}$ appears in multiple entries of $S$. Hence, to decouple such dependencies, we use a technique by Janson and Ruciński (2002), which considers the dependency graphs of the random variables and finds an equitable colouring to find independent sets of comparable sizes. To the best of our knowledge, the present work is the first study which uses the equitable colouring approach of Janson and Ruciński (2002) to derive spectral norm bounds. Ghoshdastidar et al. (2019) use this technique only to bound matrix entries.

Finally, we use concentration to show that for a sampling rate $p$ large enough, the number of comparisons ($|\mathcal{Q}|$ or $|\mathcal{T}|$) is close to its expected value. Hence, we can replace the sampling rate $p$ in the previously derived bounds by the number of comparisons, leading to differences in constants only.

**Notation.** For the sake of simplicity, we will ignore absolute constants in the inequalities stated below, and use the notations $\lesssim$ and $\gtrsim$ to write inequalities that hold up to some multiplicative absolute constant.

### C.1 Quadruplet setting

We first present the proof for the quadruplet setting using the aforementioned steps.

**Computation of $\Delta_1$.** We first derive the expectation of the AddS-4 similarity matrix $S$, where for $i \neq j$,

$$\mathbf{E}[S_{ij}] = \sum_{r \neq s} \mathbf{P}\big((i,j,r,s) \in \mathcal{Q}\big) - \mathbf{P}\big((r,s,i,j) \in \mathcal{Q}\big). \tag{10}$$

Note that the summation in AddS-4 is a sum over all distinct pairs $r, s$, noting that we do not count both $(s, r)$ and $(r, s)$ since they refer to the same comparison. To compute the expectation of each term in the summation, recall that the items belong to the planted clusters $\mathcal{C}_1, \ldots, \mathcal{C}_k$ and, for each item $i$, use $\psi_i \in [k]$ to denote the cluster index in which $i$ belongs, that is, $i \in \mathcal{C}_{\psi_i}$. The expected values of the terms are given in Table 1.

Table 1: Value of each term in the summation in (10), assuming $i \neq j$, $r \neq s$, $(i,j) \neq (r,s)$.

| Case | $\mathbf{P}\big((i,j,r,s) \in \mathcal{Q}\big)$ | $\mathbf{P}\big((r,s,i,j) \in \mathcal{Q}\big)$ | Difference |
|---|---|---|---|
| $\psi_i = \psi_j;\ \psi_r = \psi_s$ | $p/2$ | $p/2$ | $0$ |
| $\psi_i = \psi_j;\ \psi_r \neq \psi_s$ | $p(1+\epsilon\delta)/2$ | $p(1-\epsilon\delta)/2$ | $p\epsilon\delta$ |
| $\psi_i \neq \psi_j;\ \psi_r = \psi_s$ | $p(1-\epsilon\delta)/2$ | $p(1+\epsilon\delta)/2$ | $-p\epsilon\delta$ |
| $\psi_i \neq \psi_j;\ \psi_r \neq \psi_s$ | $p/2$ | $p/2$ | $0$ |

We only explain the derivation of $\mathbf{P}\big((i,j,r,s) \in \mathcal{Q}\big)$ for the case $\psi_i = \psi_j$ and $\psi_r \neq \psi_s$ as the other values are computed similarly. In this case,

$$\mathbf{P}\big((i,j,r,s) \in \mathcal{Q}\big) = \mathbf{P}\big((i,j,r,s) \in \mathcal{Q}\big|(i,j),(r,s)\text{ compared}\big)\mathbf{P}\big((i,j),(r,s)\text{ compared}\big)$$

$$= p\mathbf{P}\big((i,j,r,s) \in \mathcal{Q}\big|(i,j),(r,s)\text{ compared}\big)$$

$$= p\Big[\mathbf{P}\big((i,j,r,s) \in \mathcal{Q}\big|w_{ij} > w_{rs};\ (i,j),(r,s)\text{ compared}\big)\mathbf{P}(w_{ij} > w_{rs})$$

$$\qquad + \mathbf{P}\big((i,j,r,s) \in \mathcal{Q}\big|w_{ij} < w_{rs};\ (i,j),(r,s)\text{ compared}\big)\mathbf{P}(w_{ij} < w_{rs})\Big]$$

$$= p\left[\frac{(1+\epsilon)}{2}\frac{(1+\delta)}{2} + \frac{(1-\epsilon)}{2}\frac{(1-\delta)}{2}\right] = p\frac{(1+\epsilon\delta)}{2}\ ,$$

where in each product, the term $\mathbf{P}(w_{ij} > w_{rs})$ is computed from (3), and the other term, denoting flipped answers, follows from the quadruplet variant of (2). Based on Table 1, we have for $i, j$ such that $\psi_i = \psi_j$

$$\mathbf{E}[S_{ij}] = \sum_{(r,s):\psi_r \neq \psi_s} p\epsilon\delta = p\epsilon\delta \sum_{\ell=1}^{k} \frac{n_\ell(n - n_\ell)}{2}$$

if $i \neq j$. For $i = j$, obviously $\mathbf{E}[S_{ij}] = 0$. For $i, j$ such that $\psi_i \neq \psi_j$, we have

$$\mathbf{E}[S_{ij}] = \sum_{(r,s):\psi_r = \psi_s} -p\epsilon\delta = -p\epsilon\delta \sum_{\ell=1}^{k} \binom{n_\ell}{2}.$$

Hence, we define the ideal similarity matrix as $\widetilde{S}_{ij} = \mathbf{E}[S_{ij}]$ for $i \neq j$, and $\widetilde{S}_{ii} = p\epsilon\delta \sum_{\ell=1}^{k} \frac{n_\ell(n-n_\ell)}{2}$.

Observe that $\widetilde{S} = Z\Sigma Z^T$, where $Z \in \{0,1\}^{n \times k}$ is the assignment matrix for the planted clusters, and $\Sigma \in \mathbb{R}^{k \times k}$ such that $\Sigma_{\ell\ell} = p\epsilon\delta \sum_\ell \frac{n_\ell(n-n_\ell)}{2}$ and $\Sigma_{\ell\ell'} = -p\epsilon\delta \sum_\ell \binom{n_\ell}{2}$ for $\ell \neq \ell'$. Hence, in this case, we have

$$\Delta_1 = p\epsilon\delta \binom{n}{2}. \tag{11}$$

**Preliminary computations and definitions for concentration.** As noted earlier, $\widetilde{S}$ and $\mathbf{E}[S]$ are identical, except in the diagonal entries. Hence, we mainly have to obtain concentration of $f(S - \mathbf{E}[S])$, where $f$ is a non-negative scalar function. In the case of $\Delta_2$, $f$ denotes the maximum partial row sum, whereas $f$ is the spectral norm in the bound for $\|S - \widetilde{S}\|_2$. Define $\mathcal{W} = \{w_{ij} : i < j\}$ as the collection of random pairwise similarities. We write

$$S - \mathbf{E}[S] = \big(S - \mathbf{E}[S|\mathcal{W}]\big) + \big(\mathbf{E}[S|\mathcal{W}] - \mathbf{E}[S]\big),$$

where the first difference accounts for randomness in sampling and crowd noise, while the second difference accounts for the inherent noise in $\mathcal{W}$. This helps in separately concentrating both terms, which have different dependence structures. Formally, we perform the concentration of $f(S - \mathbf{E}[S])$ in the following way, assuming $f$ satisfies triangle inequality (which holds in the cases that we later consider).

$$
\begin{aligned}
\mathbf{P}\big(f(S - \mathbf{E}[S]) > t\big) &\leq \mathbf{P}\big(f(S - \mathbf{E}[S|\mathcal{W}]) + f(\mathbf{E}[S|\mathcal{W}] - \mathbf{E}[S]) > t\big) \\
&\leq \mathbf{P}\big(f(S - \mathbf{E}[S|\mathcal{W}]) > t/2\big) + \mathbf{P}\big(f(\mathbf{E}[S|\mathcal{W}] - \mathbf{E}[S]) > t/2\big) \\
&\leq \mathbf{E}_\mathcal{W}\big[\mathbf{P}_{\cdot|\mathcal{W}}\big(f(S - \mathbf{E}[S|\mathcal{W}]) > t/2\big)\big] + \mathbf{P}\big(f(\mathbf{E}[S|\mathcal{W}] - \mathbf{E}[S]) > t/2\big),
\end{aligned}
$$

where $\mathbf{P}_{\cdot|\mathcal{W}}$ denotes the probability over sampling and crowd noise, but conditioned on $\mathcal{W}$. In fact, we derive an uniform upper bound on the conditional probability, irrespective of $\mathcal{W}$, and hence the expectation is trivially bounded. To separately deal with the randomness in $\mathcal{W}$ and the randomness due to sampling and crowd noise, we write

$$S_{ij} = \sum_{r \neq s} \big(\mathbb{I}_{\{(i,j,r,s) \in \mathcal{Q}\}} - \mathbb{I}_{\{(r,s,i,j) \in \mathcal{Q}\}}\big) = \sum_{r<s} \xi_{ijrs}\big(\mathbb{I}_{\{w_{ij} > w_{rs}\}} - \mathbb{I}_{\{w_{ij} < w_{rs}\}}\big) \tag{12}$$

where $\xi_{ijrs} \in \{-1, 0, +1\}$ denotes whether the comparison between $(i,j)$ and $(r,s)$ is observed ($\xi_{ijrs} = 0$ if not observed), and whether the crowd response was correct ($\xi_{ijrs} = +1$) or flipped ($\xi_{ijrs} = -1$). Under our sampling and noise model,

$$\mathbf{P}(\xi_{ijrs} = 0) = 1 - p, \qquad \mathbf{P}(\xi_{ijrs} = 1) = \frac{p(1+\epsilon)}{2}, \qquad \mathbf{P}(\xi_{ijrs} = -1) = \frac{p(1-\epsilon)}{2}$$

and so, $\mathbf{E}[\xi_{ijrs}] = p\epsilon$ and $\mathrm{Var}(\xi_{ijrs}) \leq p$. Note that the set $\Xi = \{\xi_{ijrs} : i < j, r < s, (i,j) < (r,s)\}$ is a collection of independent random variables. Here, $(i,j) < (r,s)$ denotes a lexicographic ordering of tuples since we do not care about the ordering between $(i,j)$ and $(r,s)$.

In addition, recall that $F_{in}, F_{out}$ are continuous, and hence, with probability 1, any two pairwise similarities are distinct. Hence, we can write $\mathbb{I}_{\{w_{ij} > w_{rs}\}} - \mathbb{I}_{\{w_{ij} < w_{rs}\}} = 2\mathbb{I}_{\{w_{ij} > w_{rs}\}} - 1$. It is noted that $\xi_{ijrs}$ is independent of $(2\mathbb{I}_{\{w_{ij} > w_{rs}\}} - 1)$, and furthermore, the latter variable is deterministic conditioned on $\mathcal{W}$. Based on this and using the notation of $\xi_{ijrs}$, we write

$$
\begin{aligned}
S_{ij} - \mathbf{E}[S_{ij}|\mathcal{W}] &= \sum_{r<s} B_{ijrs}, &&\text{where} && B_{ijrs} = (\xi_{ijrs} - p\epsilon)\big(2\mathbb{I}_{\{w_{ij} > w_{rs}\}} - 1\big) \\
\mathbf{E}[S_{ij}|\mathcal{W}] - \mathbf{E}[S_{ij}] &= \sum_{r<s} B'_{ijrs}, &&\text{where} && B'_{ijrs} = 2p\epsilon\big(\mathbb{I}_{\{w_{ij} > w_{rs}\}} - \mathbf{P}(w_{ij} > w_{rs})\big).
\end{aligned}
\tag{13}
$$

We make the following observations about the collection of random variables $B_{ijrs}, B'_{ijrs}$, which are crucial to the subsequent concentration results. It is easy to see that $|B_{ijrs}| \leq 2$, $|B'_{ijrs}| \leq 2p\epsilon$ with probability 1, and $\mathbf{E}[B_{jrs}] = \mathbf{E}[B'_{ijrs}] = 0$, $\mathrm{Var}\,(B_{ijrs}) \leq p$ and $\mathrm{Var}\,(B'_{ijrs}) \leq 4p^2\epsilon^2$. Define the sets

$$
\begin{aligned}
\mathcal{B} &= \{B_{ijrs} \;:\; i < j, r < s, (i,j) \neq (r,s)\}, \\
\mathcal{B}' &= \{B'_{ijrs} \;:\; i < j, r < s, (i,j) \neq (r,s)\}, \\
\mathcal{B}_{i\ell} &= \{B_{ijrs} \;:\; j \in \mathcal{C}_\ell, j \neq i, r < s, (i,j) \neq (r,s)\} \qquad \text{for every } i \in [n], \ell \in [k], \\
\text{and} \quad \mathcal{B}'_{i\ell} &= \{B'_{ijrs} \;:\; j \in \mathcal{C}_\ell, j \neq i, r < s, (i,j) \neq (r,s)\} \qquad \text{for every } i \in [n], \ell \in [k].
\end{aligned}
\tag{14}
$$

Each of $\mathcal{B}$ and $\mathcal{B}'$ have $\binom{n}{2}\left(\binom{n}{2} - 1\right)$ random variables. $B_{ijrs} = -B_{rsij}$, but conditioned on $\mathcal{W}$, $B_{ijrs}$ is independent of all other variables in $\mathcal{B}$. Thus, a dependency graph on $\mathcal{B}$, conditioned on $\mathcal{W}$, has a maximum degree of 1. On the other hand, $B'_{ijrs}$ depends on all the random variables of the form $B'_{ijr's'}, B'_{i'j'rs}, B'_{r's'ij}$ and $B'_{rsi'j'}$, and so, the dependence graph for $\mathcal{B}'$ has degree smaller than $4\binom{n}{2} - 7$. Similarly, $\mathcal{B}_{i\ell}, \mathcal{B}'_{i\ell}$ have at most $n_\ell\left(\binom{n}{2} - 1\right)$ random variables. While $\mathcal{B}_{i\ell}$ has a dependency graph with degree at most 1, the dependency graph of $\mathcal{B}'_{i\ell}$ has degree at most $n_\ell + \binom{n}{2} - 3$. We now use the above discussion to derive upper bounds on $\Delta_2$ and $\|S - \widetilde{S}\|_2$.

**Upper bound for $\Delta_2$.** To derive a bound on $\Delta_2$, we first note that

$$
\Delta_2 \leq \max_{i \in [n]} \max_{\ell \in [k]} \left| \frac{1}{n_\ell} \sum_{j \in \mathcal{C}_\ell} S_{ij} - \mathbf{E}[S_{ij}] \right| + \frac{a_0}{n_{\min}}
$$

where $a_0 = \widetilde{S}_{ii}$ takes into account the fact that $\widetilde{S}$ and $\mathbf{E}[S]$ differ only in diagonal terms. In the subsequent steps, we bound the first term. For any $t > 0$, the union bound leads to

$$
\mathbf{P}\left( \max_{i \in [n]} \max_{\ell \in [k]} \left| \frac{1}{n_\ell} \sum_{j \in \mathcal{C}_\ell} S_{ij} - \mathbf{E}[S_{ij}] \right| > t \right)
$$

$$
\leq \sum_{i \in [n]} \sum_{\ell \in [k]} \mathbf{P}\left( \left| \sum_{j \in \mathcal{C}_\ell} S_{ij} - \mathbf{E}[S_{ij}|\mathcal{W}] + \mathbf{E}[S_{ij}|\mathcal{W}] - \mathbf{E}[S_{ij}] \right| > n_\ell t \right).
$$

$$
\leq \sum_{i \in [n]} \sum_{\ell \in [k]} \mathbf{E}_{\mathcal{W}}\left[ \mathbf{P}_{\cdot|\mathcal{W}}\left( \left| \sum_{j \in \mathcal{C}_\ell} S_{ij} - \mathbf{E}[S_{ij}|\mathcal{W}] \right| > \frac{n_\ell t}{2} \right) \right] + \mathbf{P}\left( \left| \sum_{j \in \mathcal{C}_\ell} \mathbf{E}[S_{ij}|\mathcal{W}] - \mathbf{E}[S_{ij}] \right| > n_\ell t \right)
$$

$$
\leq \sum_{i \in [n]} \sum_{\ell \in [k]} \mathbf{E}_{\mathcal{W}}\left[ \mathbf{P}_{\cdot|\mathcal{W}}\left( \left| \sum_{j \in \mathcal{C}_\ell} \sum_{r < s} B_{ijrs} \right| > \frac{n_\ell t}{2} \right) \right] + \mathbf{P}\left( \left| \sum_{j \in \mathcal{C}_\ell} \sum_{r < s} B'_{ijrs} \right| > \frac{n_\ell t}{2} \right) \tag{15}
$$

For the probability conditioned on $\mathcal{W}$, the summation involves terms in the set $\mathcal{B}_{i\ell}$ in (14). The discussion on $\mathcal{B}_{i\ell}$ shows that, conditioned on $\mathcal{W}$, the summation is a sum of independent random variables $B_{ijrs}$ whose properties are stated after (13). Hence, we can apply Bernstein's inequality to bound the conditional probability as

$$
\mathbf{P}_{\cdot|\mathcal{W}}\left( \left| \sum_{j \in \mathcal{C}_\ell} \sum_{r < s} B_{ijrs} \right| > \frac{n_\ell t}{2} \right) \leq 2\exp\left( -\frac{(\frac{n_\ell t}{2})^2}{2pn_\ell\binom{n}{2} + \frac{2}{3}2\frac{n_\ell t}{2}} \right)
$$

$$
\lesssim 2\exp\left( -\min\left\{ \frac{n_\ell t^2}{pn^2}, n_\ell t \right\} \right) \lesssim \frac{1}{n^3}
$$

for $t \gtrsim \max\left\{ \sqrt{\frac{pn^2 \ln n}{n_{\min}}}, \frac{\ln n}{n_{\min}} \right\}$. Since the $\mathcal{O}\left(\frac{1}{n^3}\right)$ bound on the probability holds uniformly for all $\mathcal{W}$, it also bounds the first term in (15).

For the second probability in (15), note that the $B'_{ijrs}$ in the summation are not independent, and we cannot directly apply Bernstein inequality. Hence, we apply the technique in Janson and Ruciński

(2002, Theorem 5) which bounds the probability by partitioning the random variables in $\mathcal{B}'_{i\ell}$ into independent sets. Since the dependency graph on $\mathcal{B}'_{i\ell}$ has maximum degree $d = n_\ell + \binom{n}{2} - 3$, we can obtain an equitable $(d+1)$-colouring, with each independent set of size $\lfloor |\mathcal{B}'_{i\ell}|/(d+1) \rfloor$ or $\lceil |\mathcal{B}'_{i\ell}|/(d+1) \rceil$, which are both smaller than $n_\ell$. Denote the independent sets by $\mathcal{B}'_{i\ell,(1)}, \ldots, \mathcal{B}'_{i\ell,(d+1)}$, and we can apply Bernstein's inequality to bound the summation over each independent set. Hence, we bound the second probability in (15), for every $i, \ell$, as

$$
\mathbf{P}\left( \left| \sum_{j \in \mathcal{C}_\ell} \sum_{r<s} B'_{ijrs} \right| > \frac{n_\ell t}{2} \right)
$$

$$
\leq \mathbf{P}\left( \max_{r \in \{1,\ldots,d+1\}} \left| \sum_{B' \in \mathcal{B}'_{i\ell,(r)}} B' \right| > \frac{n_\ell t}{2(d+1)} \right)
$$

$$
\leq \sum_{r=1}^{d+1} \mathbf{P}\left( \left| \sum_{B' \in \mathcal{B}'_{i\ell,(r)}} B' \right| > \frac{n_\ell t}{2(d+1)} \right) \qquad \text{(union bound)}
$$

$$
\leq 2(d+1) \exp\left( -\frac{(\frac{n_\ell t}{2(d+1)})^2}{2 \sum_{B' \in \mathcal{B}'_{i\ell,(r)}} \mathrm{Var}\left(B'\right) + \frac{2}{3} 2p\epsilon \frac{n_\ell t}{2(d+1)}} \right) \qquad \text{(Bernstein bound)}
$$

$$
\leq 2(d+1) \exp\left( -\frac{(\frac{n_\ell t}{d+1})^2}{8p^2\epsilon^2 n_\ell + \frac{2}{3} p\epsilon \frac{n_\ell t}{d+1}} \right)
$$

$$
\lesssim n^2 \exp\left( -\min\left\{ \frac{n_\ell t^2}{p^2\epsilon^2 n^4}, \frac{n_\ell t}{p\epsilon n^2} \right\} \right),
$$

which is $\mathcal{O}\left(\frac{1}{n^3}\right)$ for $t \gtrsim p\epsilon n^2 \cdot \max\left\{ \sqrt{\frac{\ln n}{n_{\min}}}, \frac{\ln n}{n_{\min}} \right\}$. The first term dominates since, under the condition on $\delta$, we have $n_{\min} \gtrsim \ln n$. Thus, we conclude that, with probability $1 - \frac{1}{4n}$,

$$
\Delta_2 \leq \max_{i \in [n]} \max_{\ell \in [k]} \left| \frac{1}{n_\ell} \sum_{j \in \mathcal{C}_\ell} S_{ij} - \mathbf{E}[S_{ij}] \right| + \frac{a_0}{n_{\min}}
$$

$$
\lesssim \max\left\{ \sqrt{\frac{pn^2 \ln n}{n_{\min}}}, \frac{\ln n}{n_{\min}}, p\epsilon n^2 \sqrt{\frac{\ln n}{n_{\min}}}, \frac{p\epsilon\delta n^2}{n_{\min}} \right\}, \qquad (16)
$$

where the last term is obviously dominated by the third term.

**Upper bound for** $\|S - \widetilde{S}\|_2$**.** Similar to the case of $\Delta_2$, we bound the spectral norm as

$$
\|S - \widetilde{S}\|_2 \leq \|S - \mathbf{E}[S|\mathcal{W}]\|_2 + \|\mathbf{E}[S|\mathcal{W}] - \mathbf{E}[S]\|_2 + \|\mathbf{E}[S] - \widetilde{S}\|_2,
$$

where the last term equals $a_0 = \widetilde{S}_{ii}$ since $\mathbf{E}[S] - \widetilde{S}$ is a diagonal matrix. For the first term, we derive a bound conditioned on $\mathcal{W}$. Recall from (13)–(14) that, conditioned on $\mathcal{W}$, the matrix $S - \mathbf{E}[S|\mathcal{W}]$ comprises of variables in $\mathcal{B}$, which has a dependence graph with degree 1. We partition $\mathcal{B}$ into two independent sets via equitable colouring, and write $S - \mathbf{E}[S|\mathcal{W}] = A + A'$, where $A$ and $A'$ are the symmetric matrices corresponding to each of the independent sets. We derive a spectral norm for each of $A$ and $A'$. For this, we first claim that, conditioned on $\mathcal{W}$, the event $\mathcal{E} = \left\{ \max_{i,j} \{|A_{ij}|, |A'_{ij}|\} \lesssim \max\left\{ \sqrt{pn^2 \ln n}, \ln n \right\} \right\}$ occurs with probability $1 - \mathcal{O}\left(\frac{1}{n}\right)$. To see this, observe that $A_{ij}$ (or $A'_{ij}$) is a sum of at most $\binom{n}{2}$ independent random variables $B_{ijrs}$. By Bernstein inequality,

$$
\mathbf{P}_{\cdot|\mathcal{W}}(|A_{ij}| > \tau) \leq 2\exp\left( -\frac{\tau^2}{2p\binom{n}{2} + \frac{4}{3}\tau} \right) \lesssim \exp\left( -\min\left\{ \frac{\tau^2}{pn^2}, \tau \right\} \right)
$$

which is $\mathcal{O}\left(\frac{1}{n^3}\right)$ for $\tau \gtrsim \max\left\{\sqrt{pn^2 \ln n}, \ln n\right\}$. Applying the union bound gives $\mathbf{P}(\mathcal{E}^c) = \mathcal{O}\left(\frac{1}{n}\right)$.

Conditioned on $\mathcal{W}$ and $\mathcal{E}$, the matrices $A, A'$ have independent zero mean entries, with each entry bounded by $\mathcal{O}\left(\max\left\{\sqrt{pn^2 \ln n}, \ln n\right\}\right)$. Furthermore, from the variance of $B_{ijrs}$, we have $\max_i \sum_j \operatorname{Var}(A_{ij}) < pn^3$, and same for $A'$. Hence, by matrix Bernstein inequality (Tropp, 2012),

$$\mathbf{P}_{\cdot|\mathcal{W},\mathcal{E}}\left(\|S - \mathbf{E}[S|\mathcal{W}]\|_2 > t\right) \leq \mathbf{P}_{\cdot|\mathcal{W},\mathcal{E}}\left(\|A\|_2 > t/2\right) + \mathbf{P}_{\cdot|\mathcal{W},\mathcal{E}}\left(\|A'\|_2 > t/2\right)$$

$$\leq 2n \exp\left(-\frac{t^2/4}{pn^3 + \frac{1}{3}t \cdot \max\left\{\sqrt{pn^2 \ln n}, \ln n\right\}}\right)$$

$$\lesssim n \exp\left(-\min\left\{\frac{t^2}{pn^3}, \frac{t}{\sqrt{pn^2 \ln n}}, \frac{t}{\ln n}\right\}\right) \lesssim \frac{1}{n}$$

for $t \gtrsim \left\{\sqrt{pn^3 \ln n}, \sqrt{pn^2(\ln n)^3}, (\ln n)^2\right\}$, where the second term is smaller than the first for $n$ large enough. Denote the complement of $\mathcal{E}$ by $\mathcal{E}^c$. For $t$ satisfying the stated condition,

$$\mathbf{P}\left(\|S - \mathbf{E}[S|\mathcal{W}]\|_2 > t\right)$$
$$= \mathbf{E}_{\mathcal{W}}\left[\mathbf{P}_{\cdot|\mathcal{W}}\left(\|S - \mathbf{E}[S|\mathcal{W}]\|_2 > t\right)\right]$$
$$= \mathbf{E}_{\mathcal{W}}\left[\mathbf{P}_{\cdot|\mathcal{W},\mathcal{E}}\left(\|S - \mathbf{E}[S|\mathcal{W}]\|_2 > t\right)\mathbf{P}_{\cdot|\mathcal{W}}(\mathcal{E}) + \mathbf{P}_{\cdot|\mathcal{W},\mathcal{E}^c}\left(\|S - \mathbf{E}[S|\mathcal{W}]\|_2 > t\right)\mathbf{P}_{\cdot|\mathcal{W}}(\mathcal{E}^c)\right]$$
$$\lesssim \mathbf{E}_{\mathcal{W}}\left[\mathbf{P}_{\cdot|\mathcal{W},\mathcal{E}}\left(\|S - \mathbf{E}[S|\mathcal{W}]\|_2 > t\right) + \mathbf{P}_{\cdot|\mathcal{W}}(\mathcal{E}^c)\right]$$
$$\lesssim \frac{1}{n}$$

as each term in the expectation is $\mathcal{O}\left(\frac{1}{n}\right)$. Thus, we have $\|S - \mathbf{E}[S|\mathcal{W}]\|_2 \lesssim \left\{\sqrt{pn^3 \ln n}, (\ln n)^2\right\}$.

To bound $\|\mathbf{E}[S|\mathcal{W}] - \mathbf{E}[S]\|_2$, we note that the entries of the matrix comprises of mutually dependent variables in the set $\mathcal{B}'$ defined in (14). We need to partition the entries into independent sets. Since the dependency graph for $\mathcal{B}'$ has maximum degree $d = 4\left(\binom{n}{2} - 1\right)$, we can partition $\mathcal{B}'$ into $d + 1$ independent sets of nearly identical sizes (equitable colouring). Let $\mathbf{E}[S|\mathcal{W}] - \mathbf{E}[S] = A^{(1)} + \ldots + A^{(d+1)}$ denote the corresponding partition of the matrix, where $A^{(\ell)} \in \mathbb{R}^{n \times n}$ is a symmetric matrix, consisting of the variables in the $\ell$-th independent set. Due to independence of variables, we have $A_{ij}^{(\ell)} = B'_{ijrs}$ for some $r, s$, and hence, we can conclude that each $A^{(\ell)}$ is a symmetric matrix with independent zero-mean entries, bounded by $2p\epsilon$ and variance at most $4p^2\epsilon^2$ (follows from properties of $B'_{ijrs}$). Thus, by matrix Bernstein inequality (Tropp, 2012), we have

$$\mathbf{P}\left(\|A^{(\ell)}\|_2 > \tau\right) \leq n \exp\left(-\frac{t^2}{8p^2\epsilon^2 n + \frac{2}{3}p\epsilon t}\right),$$

and combining with the union bound,

$$\mathbf{P}\left(\|\mathbf{E}[S|\mathcal{W}] - \mathbf{E}[S]\|_2 > t\right) \leq \mathbf{P}\left(\max_{\ell \in [d+1]} \|A^{(\ell)}\|_2 > \frac{t}{d+1}\right)$$

$$\leq n(d+1) \exp\left(-\frac{(\frac{t}{d+1})^2}{8p^2\epsilon^2 n + \frac{2}{3}p\epsilon \frac{t}{d+1}}\right)$$

$$\lesssim n^3 \exp\left(-\min\left\{\frac{t^2}{p^2\epsilon^2 n^5}, \frac{t}{p\epsilon n^2}\right\}\right),$$

which is $\mathcal{O}\left(\frac{1}{n}\right)$ for $t \gtrsim p\epsilon n^2 \cdot \max\left\{\sqrt{n \ln n}, \ln n\right\}$, where the first term obviously dominates. Combining the above derivations, we have with probability $1 - \frac{1}{4n}$,

$$\|S - \widetilde{S}\|_2 \lesssim \max\left\{\sqrt{pn^3 \ln n}, (\ln n)^2, p\epsilon n^2 \sqrt{n \ln n}, p\epsilon \delta n^2\right\} \tag{17}$$

where the last term (arising due to $a_0$) is dominated by the third.

**Deriving interval for $\lambda$ in terms of $|\mathcal{Q}|$.** We now use (11), (16) and (17) to complete the proof for the quadruplet setting. To this end, our main objective is to verify the conditions in Proposition 1:

$$\frac{\Delta_1}{2} < \Delta_1 - 6\Delta_2, \text{ that is, } \Delta_2 < \frac{\Delta_1}{12} \qquad \text{and} \qquad \|S - \widetilde{S}\|_2 < \frac{n_{\min}\Delta_1}{2}.$$

Using (11) and the bound in (16), we observe that $\Delta_2 \lesssim \Delta_1$ if

$$p \gtrsim \max\left\{\frac{\ln n}{\epsilon^2\delta^2 n^2 n_{\min}}, \frac{\ln n}{\epsilon\delta n^2 n_{\min}}\right\} \qquad \text{and} \qquad \delta \gtrsim \sqrt{\frac{\ln n}{n_{\min}}},$$

where the condition on $\delta$ arises due to the third term in the bound in (16). Similarly, comparing the bound in (17) to (11), we get that $\|S - \widetilde{S}\|_2 \lesssim n_{\min}\Delta_1$ if

$$p \gtrsim \max\left\{\frac{\ln n}{\epsilon^2\delta^2 n n_{\min}^2}, \frac{(\ln n)^2}{\epsilon\delta n^2 n_{\min}}\right\} \qquad \text{and} \qquad \delta \gtrsim \frac{\sqrt{n\ln n}}{n_{\min}},$$

where the condition on $\delta$ arises from the third bound in (17). Combining the above cases, we conclude that if

$$\delta \gtrsim \frac{\sqrt{n\ln n}}{n_{\min}} \qquad \text{and} \qquad p \gtrsim \frac{(\ln n)^2}{\epsilon^2\delta^2 n n_{\min}^2}, \qquad (18)$$

then the criteria for $\Delta_2$ and $\|S - \widetilde{S}\|_2$ are satisfied, and by Proposition 1, $X^*$ is the unique optimal solution for SDP-$\lambda$ with the range of $\lambda$ given by

$$\|S - \widetilde{S}\|_2 \lesssim \max\left\{\sqrt{pn^3\ln n}, p\epsilon\sqrt{n^5\ln n}, (\ln n)^2\right\} \lesssim \lambda < \frac{p\epsilon\delta n_{\min}}{2}\binom{n}{2} = \frac{\Delta_1}{2}. \qquad (19)$$

We finally show that the condition on $p$ holds under the stated condition of $|\mathcal{Q}| \gtrsim \frac{n^3(\ln n)^2}{\epsilon^2\delta^2 n_{\min}^2}$, and state the above interval for $\lambda$ in terms of $|\mathcal{Q}|$. Under the assumption that each quadruplet is observed independently with probability $p$, we have that $\mathbf{E}[|\mathcal{Q}|] = p\binom{n}{2} = O(pn^4)$. By Bernstein inequality, it is easy to verify that for $p \gtrsim \frac{\ln n}{n^4}$ or equivalently $|\mathcal{Q}| \gtrsim \ln n$, we have $|\mathcal{Q}| \in \left(\frac{1}{2}\mathbf{E}[|\mathcal{Q}|], \frac{3}{2}\mathbf{E}[|\mathcal{Q}|]\right)$ with probability $1 - \mathcal{O}\left(\frac{1}{n}\right)$. Hence, we can replace $p$ by $\frac{|\mathcal{Q}|}{n^4}$ in (18)–(19) up to difference in constants, which leads to the statement of Theorem 1 in the quadruplet setting.

## C.2 Triplet setting

The proof structure in the triplet case is similar to that of the quadruplet setting. We derive an appropriate ideal matrix $\widetilde{S}$, where $\widetilde{S} = \mathbf{E}[S]$, except for some differences in the diagonal entries since $S_{ii} = 0$ for all $i$. From the block structure of $\widetilde{S}$, we can compute $\Delta_1$. Subsequently, concentration inequalities are used to derive upper bounds on $\|S - \widetilde{S}\|_2$ and $\Delta_2$ in terms of the model parameters. As done in the analysis of AddS-4, we let $\mathcal{W}$ denote the collection of random pairwise similarities, and decompose

$$S - \widetilde{S} = \left(S - \mathbf{E}[S|\mathcal{W}]\right) + \left(\mathbf{E}[S|\mathcal{W}] - \mathbf{E}[S]\right) + \left(\mathbf{E}[S] - \widetilde{S}\right).$$

The last term is easy to tackle, and we use separate concentration for the first two terms both in the context of $\Delta_2$ and the spectral norm. Bounds on these terms, combined with Proposition 1, provide sufficient conditions on $\delta$ and sampling rate $p$ such that exact recovery occurs. Finally, we show that for $p$ large enough, the number of triplets $|\mathcal{T}|$ is close to its expected value $pn\binom{n-1}{2}$, and state the conditions in terms of $|\mathcal{T}|$.

**Computation of $\Delta_1$.** The expectation of the AddS-3 similarity $S_{ij}$, for $i \neq j$, is given by

$$\mathbf{E}[S_{ij}] = \sum_{r \neq i,j} \mathbf{P}\big((i,j,r) \in \mathcal{T}\big) - \mathbf{P}\big((i,r,j) \in \mathcal{T}\big) + \mathbf{P}\big((j,i,r) \in \mathcal{T}\big) - \mathbf{P}\big((j,r,i) \in \mathcal{T}\big). \qquad (20)$$

We now compute each term in the summation using the notation $\psi_i \in [k]$ to indicate $i \in \mathcal{C}_{\psi_i}$. The expected values of the terms are given in Table 2, where the last column represents the overall term

Table 2: Value of each term in the summation in (20), assuming $i, j, r$ are distinct.

| Case | $\mathbf{P}\big((i,j,r) \in \mathcal{T}\big)$ | $\mathbf{P}\big((i,r,j) \in \mathcal{T}\big)$ | $\mathbf{P}\big((j,i,r) \in \mathcal{T}\big)$ | $\mathbf{P}\big((j,r,i) \in \mathcal{T}\big)$ | Aggregate |
|---|---|---|---|---|---|
| $\psi_i = \psi_j = \psi_r$ | $p/2$ | $p/2$ | $p/2$ | $p/2$ | $0$ |
| $\psi_i = \psi_j \neq \psi_r$ | $p(1+\epsilon\delta)/2$ | $p(1-\epsilon\delta)/2$ | $p(1+\epsilon\delta)/2$ | $p(1-\epsilon\delta)/2$ | $2p\epsilon\delta$ |
| $\psi_i \neq \psi_j = \psi_r$ | $p/2$ | $p/2$ | $p(1-\epsilon\delta)/2$ | $p(1+\epsilon\delta)/2$ | $-p\epsilon\delta$ |
| $\psi_i = \psi_r \neq \psi_j$ | $p(1-\epsilon\delta)/2$ | $p(1+\epsilon\delta)/2$ | $p/2$ | $p/2$ | $-p\epsilon\delta$ |
| $\psi_i \neq \psi_j \neq \psi_r$ | $p/2$ | $p/2$ | $p/2$ | $p/2$ | $0$ |

for each $r \neq i, j$ in (20). The derivation for these values is identical to the one in the quadruplet setting.

Based on Table 2, we can infer that for $i \neq j$ such that $\psi_i = \psi_j$,

$$\mathbf{E}[S_{ij}] = \sum_{r \notin \mathcal{C}_{\psi_i}} 2p\epsilon\delta = 2p\epsilon\delta(n - n_{\psi_i})$$

For $i, j$ such that $\psi_i \neq \psi_j$, we have

$$\mathbf{E}[S_{ij}] = \sum_{r \in \mathcal{C}_{\psi_i}, r \neq i} (-p\epsilon\delta) + \sum_{r \in \mathcal{C}_{\psi_j}, r \neq j} (-p\epsilon\delta) = -p\epsilon\delta(n_{\psi_i} + n_{\psi_j} - 2).$$

Hence, we define the ideal similarity matrix as $\widetilde{S}_{ij} = Z\Sigma Z^T$, where $\Sigma_{\ell\ell} = 2p\epsilon\delta(n - n_\ell)$ and $\Sigma_{\ell\ell'} = -p\epsilon\delta(n_\ell + n_{\ell'} - 2)$ for $\ell \neq \ell'$, and we can compute

$$\Delta_1 = p\epsilon\delta(n - 2). \tag{21}$$

**Preliminary computations and definitions for concentration.** We define $\mathcal{W} = \{w_{ij} : i < j\}$ as the collection of random pairwise similarities, and split the concentration of $\Delta_2$ and $\|S - \widetilde{S}\|_2$ into terms involving $S - \mathbf{E}[S|\mathcal{W}]$ and $\mathbf{E}[S|\mathcal{W}] - \mathbf{E}[S]$. The basic idea is discussed in the corresponding part of the quadruplet setting, and here, we introduce the key random variables. We first write

$$S_{ij} = \sum_{r \neq i, j} \big(\mathbb{I}_{\{(i,j,r)\in\mathcal{T}\}} - \mathbb{I}_{\{(i,r,j)\in\mathcal{T}\}}\big) + \big(\mathbb{I}_{\{(j,i,r)\in\mathcal{T}\}} - \mathbb{I}_{\{(j,r,i)\in\mathcal{T}\}}\big)$$

$$= \sum_{r \neq i, j} \xi_{ijr}\big(\mathbb{I}_{\{w_{ij}>w_{ir}\}} - \mathbb{I}_{\{w_{ij}<w_{ir}\}}\big) + \xi_{jir}\big(\mathbb{I}_{\{w_{ji}>w_{jr}\}} - \mathbb{I}_{\{w_{ji}<w_{jr}\}}\big) \tag{22}$$

where $\xi_{ijr} \in \{-1, 0, +1\}$ denotes whether the comparison between $(i, j)$ and $(i, r)$ is observed ($\xi_{ijr} = 0$ if not observed), and whether the crowd response was correct ($\xi_{ijr} = +1$) or flipped ($\xi_{ijr} = -1$). Under our sampling and noise model,

$$\mathbf{P}(\xi_{ijr} = 0) = 1 - p, \qquad \mathbf{P}(\xi_{ijr} = 1) = \frac{p(1+\epsilon)}{2}, \qquad \mathbf{P}(\xi_{ijr} = -1) = \frac{p(1-\epsilon)}{2}$$

and so, $\mathbf{E}[\xi_{ijr}] = p\epsilon$ and $\mathrm{Var}(\xi_{ijr}) \leq p$. The set $\Xi = \{\xi_{ijr} : j < r, i \neq j, r\}$ denotes the collection of such random variables, where we abuse notation by using $\xi_{ijr}$ and $\xi_{irj}$ to refer to the same variable. We note that the variables in $\Xi$ are mutually independent.

We use the continuous nature of $F_{in}, F_{out}$ to write $\mathbb{I}_{\{w_{ij}>w_{ir}\}} - \mathbb{I}_{\{w_{ij}<w_{ir}\}} = 2\mathbb{I}_{\{w_{ij}>w_{ir}\}} - 1$, and further define

$$S_{ij} - \mathbf{E}[S_{ij}|\mathcal{W}] = \sum_{r \neq i, j} B_{ijr} + B_{jir} \qquad \text{with} \quad B_{ijr} = (\xi_{ijr} - p\epsilon)\big(2\mathbb{I}_{\{w_{ij}>w_{ir}\}} - 1\big),$$

$$\mathbf{E}[S_{ij}|\mathcal{W}] - \mathbf{E}[S_{ij}] = \sum_{r \neq i, j} B'_{ijr} + B'_{jir} \quad \text{with} \quad B'_{ijr} = 2p\epsilon\big(\mathbb{I}_{\{w_{ij}>w_{ir}\}} - \mathbf{P}(w_{ij} > w_{ir})\big). \tag{23}$$

The random variables $B_{ijr}, B'_{ijr}$ have the following properties: $|B_{ijr}| \leq 2$, $|B'_{ijr}| \leq 2p\epsilon$ with probability 1, $\mathbf{E}[B_{ijr}] = \mathbf{E}[B'_{ijr}] = 0$, $\mathrm{Var}(B_{ijr}) \leq p$ and $\mathrm{Var}(B'_{ijr}) \leq 4p^2\epsilon^2$. We define the sets

$$\mathcal{B} = \{B_{ijr} : j \neq i, r \neq i, j\},$$
$$\mathcal{B}' = \{B'_{ijr} : j \neq i, r \neq i, j\},$$
$$\mathcal{B}_{i\ell} = \{B_{ijr}, B_{jir} : j \in \mathcal{C}_\ell, r \neq i, j\} \qquad \text{for every } i \in [n], \ell \in [k], \tag{24}$$
$$\text{and} \qquad \mathcal{B}'_{i\ell} = \{B'_{ijr}, B'_{jir} : j \in \mathcal{C}_\ell, r \neq i, j\} \qquad \text{for every } i \in [n], \ell \in [k].$$

Each of $\mathcal{B}$ and $\mathcal{B}'$ have $n(n-1)(n-2)$ random variables, whereas $\mathcal{B}_{i\ell}, \mathcal{B}'_{i\ell}$ have at most $2n_\ell(n-2)$ random variables. Note that $B_{ijr} = -B_{irj}$, but conditioned on $\mathcal{W}$, $B_{ijr}$ is independent of all other variables in $\mathcal{B}$. Thus, a dependency graph on $\mathcal{B}$, conditioned on $\mathcal{W}$, has a maximum degree of 1. The same is also true for $\mathcal{B}'_{i\ell}$. On the other hand, $B'_{ijr}$ depends on the random variables that involve either $w_{ij}$ or $w_{ir}$, that is $B'_{irj}, B'_{jir}, B'_{jri}, B'_{rij}, B'_{rji}$, as well as all six variants of variables $B'_{ij'r'}$ and $B'_{ij'r}, j', r' \notin \{i, j, r\}$. Thus each $B'_{ijr}$ depends on at most $5 + 12(n-3) = \mathcal{O}(n)$ variables in $\mathcal{B}'$. The same holds when we restrict the set to $\mathcal{B}'_{i\ell}$. Thus, the dependency graph for $\mathcal{B}'$ and $\mathcal{B}'_{i\ell}$ have dependency graph with $\mathcal{O}(n)$ maximum degree. We now use the above defined random variables and their properties to derive upper bounds on $\Delta_2$ and $\|S - \widetilde{S}\|_2$.

**Upper bound for $\Delta_2$.** To derive a bound on $\Delta_2$, we first note that

$$\Delta_2 \leq \max_{i \in [n]} \max_{\ell \in [k]} \left| \frac{1}{n_\ell} \sum_{j \in \mathcal{C}_\ell} S_{ij} - \mathbf{E}[S_{ij}|\mathcal{W}] \right| + \max_{i \in [n]} \max_{\ell \in [k]} \left| \frac{1}{n_\ell} \sum_{j \in \mathcal{C}_\ell} \mathbf{E}[S_{ij}|\mathcal{W}] - \mathbf{E}[S_{ij}] \right| + \max_{i \in [n]} \frac{\widetilde{S}_{ii}}{n_{\min}}.$$

In the subsequent steps, we bound the first term. For any $t > 0$, the union bound leads to

$$\mathbf{P}\left( \max_{i \in [n]} \max_{\ell \in [k]} \left| \frac{1}{n_\ell} \sum_{j \in \mathcal{C}_\ell} S_{ij} - \mathbf{E}[S_{ij}|\mathcal{W}] \right| > t \right)$$

$$\leq \sum_{i \in [n]} \sum_{\ell \in [k]} \mathbf{P}\left( \left| \sum_{j \in \mathcal{C}_\ell} S_{ij} - \mathbf{E}[S_{ij}|\mathcal{W}] \right| > n_\ell t \right)$$

$$= \sum_{i \in [n]} \sum_{\ell \in [k]} \mathbf{E}_\mathcal{W} \left[ \mathbf{P}_{\cdot|\mathcal{W}}\left( \left| \sum_{j \in \mathcal{C}_\ell} \sum_{r \neq i,j} B_{ijr} + B_{jir} \right| > n_\ell t \right) \right]$$

The summation inside the conditional probability involves terms in $\mathcal{B}_{i\ell}$ defined in (24), and the previous discussions show that the dependency graph of $\mathcal{B}_{i\ell}$ has maximum degree of 1. Hence, we can split the $2n_\ell(n-2)$ variables in $\mathcal{B}_{i\ell}$ into two independent sets, say $\mathcal{B}_{i\ell,(1)}$ and $\mathcal{B}_{i\ell,(2)}$, and derive concentration for each of them separately using Bernstein inequality in the following way.

$$\mathbf{P}\left( \max_{i \in [n]} \max_{\ell \in [k]} \left| \frac{1}{n_\ell} \sum_{j \in \mathcal{C}_\ell} S_{ij} - \mathbf{E}[S_{ij}|\mathcal{W}] \right| > t \right)$$

$$\leq \sum_{i \in [n]} \sum_{\ell \in [k]} \mathbf{E}_\mathcal{W} \left[ \mathbf{P}_{\cdot|\mathcal{W}}\left( \left| \sum_{B \in \mathcal{B}_{i\ell,(1)}} B \right| > \frac{n_\ell t}{2} \right) + \left[ \mathbf{P}_{\cdot|\mathcal{W}}\left( \left| \sum_{B \in \mathcal{B}_{i\ell,(2)}} B \right| > \frac{n_\ell t}{2} \right) \right] \right]$$

$$\leq \sum_{i \in [n]} \sum_{\ell \in [k]} \mathbf{E}_\mathcal{W} \left[ 2\exp\left( -\frac{(n_\ell t/2)^2}{2p|\mathcal{B}_{i\ell,(1)}| + 2n_\ell t/3} \right) + 2\exp\left( -\frac{(n_\ell t/2)^2}{2p|\mathcal{B}_{i\ell,(2)}| + 2n_\ell t/3} \right) \right]$$

$$\lesssim n^2 \exp\left( -\min\left\{ \frac{n_{\min} t^2}{pn}, n_{\min} t \right\} \right),$$

where the last step follows by noting that each of the two independent sets have $\Omega(nn_\ell)$ variables, and the bounds are independent of $\mathcal{W}$. The above probability is $\mathcal{O}\left( \frac{1}{n} \right)$ for $t \gtrsim \max\left\{ \sqrt{\frac{pn \ln n}{n_{\min}}}, \frac{\ln n}{n_{\min}} \right\}$.

For the second term in the upper bound for $\Delta_2$, we have

$$\mathbf{P}\left( \max_{i \in [n]} \max_{\ell \in [k]} \left| \frac{1}{n_\ell} \sum_{j \in \mathcal{C}_\ell} \mathbf{E}[S_{ij}|\mathcal{W}] - \mathbf{E}[S_{ij}] \right| > t \right) \leq \sum_{i \in [n]} \sum_{\ell \in [k]} \mathbf{P}\left( \left| \sum_{j \in \mathcal{C}_\ell} \sum_{r \neq i,j} B'_{ijr} + B'_{jir} \right| > n_\ell t \right)$$

where the tail bound is for the sum of all random variables in $\mathcal{B}'_{i\ell}$. Since the dependency graph on $\mathcal{B}'_{i\ell}$ has maximum degree $d = \mathcal{O}(n)$, we can obtain an equitable $(d+1)$-colouring with each independent set of size $\lfloor |\mathcal{B}'_{i\ell}|/(d+1) \rfloor$ or $\lceil |\mathcal{B}'_{i\ell}|/(d+1) \rceil$, which are smaller than $n_\ell$. We denote the

independent sets by $\mathcal{B}'_{i\ell,(1)}, \ldots, \mathcal{B}'_{i\ell,(d+1)}$, and use Bernstein inequality to bound the summation over each independent set. Hence, we bound the probability for every $i, \ell$, as

$$\mathbf{P}\left(\left|\sum_{j \in \mathcal{C}_\ell} \sum_{r \neq i,j} B'_{ijr} + B'_{jir}\right| > n_\ell t\right) \leq \mathbf{P}\left(\max_{r \in \{1, \ldots, d+1\}} \left|\sum_{B' \in \mathcal{B}'_{i\ell,(r)}} B'\right| > \frac{n_\ell t}{(d+1)}\right)$$

$$\leq \sum_{r=1}^{d+1} \mathbf{P}\left(\left|\sum_{B' \in \mathcal{B}'_{i\ell,(r)}} B'\right| > \frac{n_\ell t}{(d+1)}\right)$$

$$\leq 2(d+1) \exp\left(-\frac{(\frac{n_\ell t}{(d+1)})^2}{8p^2\epsilon^2 |\mathcal{B}'_{i\ell,(r)}| + \frac{4}{3}p\epsilon \frac{n_\ell t}{(d+1)}}\right)$$

$$\lesssim n \exp\left(-\min\left\{\frac{n_\ell t^2}{p^2\epsilon^2 n^2}, \frac{n_\ell t}{p\epsilon n}\right\}\right),$$

which is $\mathcal{O}\left(\frac{1}{n^3}\right)$ for $t \gtrsim p\epsilon n \cdot \max\left\{\sqrt{\frac{\ln n}{n_{\min}}}, \frac{\ln n}{n_{\min}}\right\}$. The first term dominates for $n_{\min} \gtrsim \ln n$, which arises due to the condition on $\delta$. Combining the above discussions we claim that, with probability $1 - \frac{1}{4n}$,

$$\Delta_2 \lesssim \max\left\{\sqrt{\frac{pn \ln n}{n_{\min}}}, \frac{\ln n}{n_{\min}}, p\epsilon n \sqrt{\frac{\ln n}{n_{\min}}}, \frac{p\epsilon \delta n}{n_{\min}}\right\}, \tag{25}$$

where the last term is obviously dominated by the third term.

**Upper bound for $\|S - \widetilde{S}\|_2$.** Similar to the case of $\Delta_2$, we bound

$$\|S - \widetilde{S}\|_2 \leq \|S - \mathbf{E}[S|\mathcal{W}]\|_2 + \|\mathbf{E}[S|\mathcal{W}] - \mathbf{E}[S]\|_2 + \|\mathbf{E}[S] - \widetilde{S}\|_2,$$

where the last term equals $\max_i \widetilde{S}_{ii}$. For the first term, we derive a bound conditioned on $\mathcal{W}$. Recall from (23)–(24) that, conditioned on $\mathcal{W}$, the matrix $S - \mathbf{E}[S|\mathcal{W}]$ comprises of variables in $\mathcal{B}$, which has a dependence graph with degree 1. We partition $\mathcal{B}$ into two independent sets via equitable colouring, and write $S - \mathbf{E}[S|\mathcal{W}] = A + A'$, where $A$ and $A'$ are the matrices corresponding to each of the independent sets. We derive a spectral norm for each of $A$ and $A'$. For this, we first claim that, conditioned on $\mathcal{W}$, the event $\mathcal{E} = \left\{\max_{i,j}\left\{|A_{ij}|, |A'_{ij}|\right\} \lesssim \max\left\{\sqrt{pn \ln n}, \ln n\right\}\right\}$ occurs with probability $1 - \mathcal{O}\left(\frac{1}{n}\right)$. To see this, observe that $A_{ij}$ (or $A'_{ij}$) is a sum of $2(n-2)$ independent random variables $B_{ijr}, B_{jir}$. By Bernstein inequality,

$$\mathbf{P}_{\cdot|\mathcal{W}}(|A_{ij}| > \tau) \leq 2\exp\left(-\frac{\tau^2}{4p(n-2) + \frac{4}{3}\tau}\right) \lesssim \exp\left(-\min\left\{\frac{\tau^2}{pn}, \tau\right\}\right)$$

which is $\mathcal{O}\left(\frac{1}{n^3}\right)$ for $\tau \gtrsim \max\left\{\sqrt{pn \ln n}, \ln n\right\}$. Applying the union bound gives $\mathbf{P}(\mathcal{E}^c) = \mathcal{O}\left(\frac{1}{n}\right)$.

Conditioned on $\mathcal{W}$ and $\mathcal{E}$, the matrices $A, A'$ have independent zero mean entries, with each entry bounded by $\mathcal{O}\left(\max\left\{\sqrt{pn \ln n}, \ln n\right\}\right)$. Furthermore, from the variance of $B_{ijr}$, we have $\max_i \sum_j \text{Var}\left(A_{ij}\right) < 2pn^2$, and the same holds for $A'$. Hence, by matrix Bernstein inequality (Tropp, 2012),

$$\mathbf{P}_{\cdot|\mathcal{W},\mathcal{E}}\left(\|S - \mathbf{E}[S|\mathcal{W}]\|_2 > t\right) \leq \mathbf{P}_{\cdot|\mathcal{W},\mathcal{E}}\left(\|A\|_2 > t/2\right) + \mathbf{P}_{\cdot|\mathcal{W},\mathcal{E}}\left(\|A'\|_2 > t/2\right)$$

$$\leq 2n \exp\left(-\frac{t^2/4}{pn^2 + \frac{1}{3}t \cdot \max\left\{\sqrt{pn \ln n}, \ln n\right\}}\right)$$

$$\lesssim n \exp\left(-\min\left\{\frac{t^2}{pn^2}, \frac{t}{\sqrt{pn \ln n}}, \frac{t}{\ln n}\right\}\right) \lesssim \frac{1}{n}$$

for $t \gtrsim \left\{\sqrt{pn^2 \ln n}, \sqrt{pn(\ln n)^3}, (\ln n)^2\right\}$, where the second term is smaller than the first for $n$ large enough. As in the quadruplet setting, we add the probability $\mathbf{P}(\mathcal{E}^c)$ and take expectation over $\mathcal{W}$ to obtain $\|S - \mathbf{E}[S|\mathcal{W}]\|_2 \lesssim \left\{\sqrt{pn^2 \ln n}, (\ln n)^2\right\}$ with probability $1 - \mathcal{O}\left(\frac{1}{n}\right)$.

To bound $\|\mathbf{E}[S|\mathcal{W}] - \mathbf{E}[S]\|_2$, we note that the entries of the matrix comprises of mutually dependent variables in the set $\mathcal{B}'$ defined in (24). Since the dependency graph for $\mathcal{B}'$ has maximum degree $d = \mathcal{O}(n)$, we partition $\mathcal{B}'$ into $d+1$ independent sets of nearly identical sizes (equitable colouring). Let $\mathbf{E}[S|\mathcal{W}] - \mathbf{E}[S] = A^{(1)} + \ldots + A^{(d+1)}$ denote the corresponding partition of the matrix, where $A^{(\ell)} \in \mathbb{R}^{n \times n}$ is a symmetric matrix consisting of the variables in the $\ell$-th independent set. Due to the independence of the variables, we have $A^{(\ell)}_{ij} = A^{(\ell)}_{ji} = B'_{ijr}$ or $B'_{jir}$ for some $r \neq i, j$. Hence, each $A^{(\ell)}$ is a symmetric matrix with independent zero-mean entries, bounded by $2p\epsilon$ and variance at most $4p^2\epsilon^2$ (follows from properties of $B'_{ijr}$). Thus, by matrix Bernstein inequality (Tropp, 2012), we have

$$\mathbf{P}\left(\|A^{(\ell)}\|_2 > \tau\right) \leq n \exp\left(-\frac{t^2}{8p^2\epsilon^2 n + \frac{2}{3}p\epsilon t}\right),$$

and combining with the union bound,

$$\mathbf{P}\left(\|\mathbf{E}[S|\mathcal{W}] - \mathbf{E}[S]\|_2 > t\right) \leq \mathbf{P}\left(\max_{\ell \in [d+1]} \|A^{(\ell)}\|_2 > \frac{t}{d+1}\right)$$

$$\leq n(d+1)\exp\left(-\frac{(\frac{t}{d+1})^2}{8p^2\epsilon^2 n + \frac{2}{3}p\epsilon\frac{t}{d+1}}\right)$$

$$\lesssim n^2 \exp\left(-\min\left\{\frac{t^2}{p^2\epsilon^2 n^3}, \frac{t}{p\epsilon n}\right\}\right),$$

which is $\mathcal{O}\left(\frac{1}{n}\right)$ for $t \gtrsim p\epsilon n \cdot \max\left\{\sqrt{n \ln n}, \ln n\right\}$, where the first term obviously dominates. Combining the above derivations, we have with probability $1 - \frac{1}{4n}$,

$$\|S - \widetilde{S}\|_2 \lesssim \max\left\{\sqrt{pn^2 \ln n}, (\ln n)^2, p\epsilon n\sqrt{n \ln n}, p\epsilon\delta n\right\} \tag{26}$$

where the last term (arising due to $\max_i \widetilde{S}_{ii}$) is dominated by the third.

**Deriving interval for $\lambda$ in terms of $|\mathcal{T}|$.** We now use (21), (25) and (26) to complete the proof for the triplet setting. We verify the conditions in Proposition 1 by deriving conditions under which $\Delta_2 < \frac{1}{12}\Delta_1$ and $\|S - \widetilde{S}\|_2 < \frac{1}{2}n_{\min}\Delta_1$. Similar to the proof for the quadruplet setting, we compare the upper bounds in (16) and (17) with $\Delta_1$ and $n_{\min}\Delta_1$, respectively. As in the previous setting, the first two bounds in (16)–(17) lead to conditions on $p$, while the third term leads to a condition on $\delta$. Combining the different cases, it follows that if

$$\delta \gtrsim \frac{\sqrt{n \ln n}}{n_{\min}} \qquad \text{and} \qquad p \gtrsim \frac{(\ln n)^2}{\epsilon^2 \delta^2 n_{\min}^2}, \tag{27}$$

then the criteria for $\Delta_2$ and $\|S - \widetilde{S}\|_2$ are satisfied, and by Proposition 1, $X^*$ is the unique optimal solution for SDP-$\lambda$ with the range of $\lambda$ given by

$$\|S - \widetilde{S}\|_2 \lesssim \max\left\{\sqrt{pn^2 \ln n}, p\epsilon\sqrt{n^3 \ln n}, (\ln n)^2\right\} \lesssim \lambda < \frac{p\epsilon\delta n_{\min}(n-2)}{2} = \frac{\Delta_1}{2}. \tag{28}$$

We finally show that the condition on $p$ holds under the stated condition of $|\mathcal{T}| \gtrsim \frac{n^3(\ln n)^2}{\epsilon^2 \delta^2 n_{\min}^2}$, and state the above interval for $\lambda$ in terms of $|\mathcal{T}|$. Under the assumption that each triplet is observed independently with probability $p$, we $\mathbf{E}[|\mathcal{T}|] = pn\binom{n-1}{2} = \mathcal{O}\left(pn^3\right)$. By Bernstein inequality, it is easy to verify that for $p \gtrsim \frac{\ln n}{n^3}$ or equivalently $|\mathcal{T}| \gtrsim \ln n$, we have $|\mathcal{T}| \in \left(\frac{1}{2}\mathbf{E}[|\mathcal{T}|], \frac{3}{2}\mathbf{E}[|\mathcal{T}|]\right)$ with probability $1 - \mathcal{O}\left(\frac{1}{n}\right)$. Hence, we can replace $p$ by $\frac{|\mathcal{T}|}{n^3}$ in (27)–(28) up to differences in constants, which leads to the statement of Theorem 1 in the triplet setting.

# D  Algorithmic details

In this section, we provide details on the modified SPUR algorithm that we use to tune the parameter $\lambda$, and to select the number of clusters.

SPUR, acronym for Semidefinite Program with Unknown $r$ ($r$ denoting the number of clusters), was proposed by Yan et al. (2018) to tune the parameter $\lambda$ of SDP-$\lambda$ in the context of graph clustering (see Algorithm 2). The underlying idea of this approach is to search for the optimal $\lambda$ using a grid search over the range $0 < \lambda < \lambda_{\max}$, where $\lambda_{\max}$ is derived from an exact recovery result under stochastic block model.

---

**Algorithm 2:** Semidefinite Program with Unknown $k$ (SPUR).

---

**input** : graph $A$, number of candidates $T$.
**begin**
    **for** $t = 1$ *to* $T$ **do**
        $\lambda_t = \exp\left(\frac{t}{T}\ln\left(1 + \lambda_{\max}\right)\right) - 1.$       (Yan et al. (2018) set $\lambda_{\max} = \|A\|_{op}$)
        Solve SDP-$\lambda$ with $\lambda = \lambda_t$ to obtain $X_t$.
        Estimate $k_t = $ integer approximation of trace $(X_t)$.
    **end**
    Choose $\hat{t} = \arg\max\limits_{t} \dfrac{\sum_{i \leq k_t} \sigma_i(X_t)}{\text{trace}\,(X_t)}$, where $\sigma_i(X_t)$ denotes $i$-th largest eigenvalue of $X_t$.
**end**
**output** : Number of clusters $k_{\hat{t}}$, $X_{\hat{t}}$.

---

In the present setting, Theorem 1 shows that the planted clusters can be exactly recovered given a sufficient number of comparisons and an appropriate choice of $\lambda$. From Theorem 1, a candidate for $\lambda_{\max}$ can be chosen as $\frac{|\mathcal{T}|}{n}$ (for triplets) or $\frac{|\mathcal{Q}|}{n}$ (for quadruplets), which is a loose upper bound for the theoretical interval for $\lambda$, obtained by noting that $\epsilon\delta n_{\min} \leq n$. Thus, following Yan et al. (2018), we could use Algorithm 2 with our choice of $\lambda_{\max}$.

Unfortunately, this approach has two main drawbacks. First, it ignores the lower bound in Theorem 1 and, second, setting $T$, the number of $\lambda$ values that should be considered in Algorithm 2, is difficult. To address the former issue, we propose to consider Theorem 1 once more and to use $\lambda_{\min} = \sqrt{c(\ln n)/n}$ as a lower bound for $\lambda$ instead of 0, as used in Yan et al. (2018). To address the latter issue, we use the fact that the estimated number of clusters $k$ monotonically decreases with $\lambda$ as shown in the next Lemma.

**Lemma 1** (**The estimated number of clusters decreases monotonically with increasing** $\lambda$). *For any $\lambda > 0$, let $X_\lambda$ denote the solution of SDP-$\lambda$ and $k_\lambda = \lfloor\text{trace}\,(X_\lambda)\rceil$ be the integer approximation of trace $(X_\lambda)$, which is an estimate of the number of clusters. Then, $k_\lambda$ is a non-increasing function of $\lambda$, that is*

$$\lambda' \geq \lambda \Rightarrow k_{\lambda'} \leq k_\lambda.$$

*Proof.* We start this proof by noting that since $k_\lambda$ is the integer approximation of trace $(X_\lambda)$, it suffices to show that trace $(X_\lambda)$ is a non-increasing function of $\lambda$. Then, consider distinct $\lambda', \lambda$ and let $X_{\lambda'}, X_\lambda$ be the solutions of SDP-$\lambda$ with parameters $\lambda', \lambda$, respectively. We have

$$\text{trace}\,(SX_\lambda) - \lambda\text{trace}\,(X_\lambda) \geq \text{trace}\,(SX_{\lambda'}) - \lambda\text{trace}\,(X_{\lambda'})\ ,$$
$$\text{trace}\,(SX_\lambda) - \lambda'\text{trace}\,(X_\lambda) \leq \text{trace}\,(SX_{\lambda'}) - \lambda'\text{trace}\,(X_{\lambda'})\ .$$

Subtracting the second inequality from the first inequality implies

$$\text{trace}\,(SX_\lambda) - \lambda\text{trace}\,(X_\lambda) - \left(\text{trace}\,(SX_\lambda) - \lambda'\text{trace}\,(X_\lambda)\right)$$
$$\geq \text{trace}\,(SX_{\lambda'}) - \lambda\text{trace}\,(X_{\lambda'}) - \text{trace}\,(SX_{\lambda'}) + \lambda'\text{trace}\,(X_{\lambda'})$$

which implies

$$(\lambda' - \lambda)\text{trace}\,(X_\lambda) \geq (\lambda' - \lambda)\text{trace}\,(X_{\lambda'})$$

or equivalently, $(\lambda' - \lambda)(\text{trace}\,(X_{\lambda'}) - \text{trace}\,(X_\lambda)) \leq 0$. Thus, for $\lambda' > \lambda$, we can conclude that trace $(X_{\lambda'}) \leq$ trace $(X_\lambda)$, which shows that trace $(X_\lambda)$ and $k_\lambda$ are non-increasing functions of $\lambda$. $\square$

Following this, using $\lambda_{\min}$ and $\lambda_{\max}$, we get two estimates of the number of clusters, $k_{\lambda_{\min}}$ and $k_{\lambda_{\max}}$. Then, we search over $k \in [k_{\lambda_{\max}}, k_{\lambda_{\min}}]$ instead of searching over $\lambda$—in practice, it helps to search over the values $\max\{2, k_{\lambda_{\max}}\} \leq k \leq k_{\lambda_{\min}} + 2$. We select $k$ that maximises the above SPUR objective, where $X_k$ is computed using the simpler SDP-$k$ (Yan et al., 2018). This approach is summarized in Algorithm 1 in the main paper.

Figure 3: Comparing clustering algorithms to partition $X$ in the last step. Using k-means or spectral clustering does not affect the output of our approach.

# E    Additional results for the planted model

In this section, we provide additional experiments on our planted model. We show that changing the clustering method used in the last step of our approach to cluster the matrix $X$ learned by SDP-$\lambda$ or SDP-$k$ does not affect the results. We demonstrate that, given a sufficient number of comparisons, SPUR correctly estimates the number of clusters. We give details on the distributions used in Figure 1c. Finally, we consider several additional experiments where we vary the planted model parameters that were ignored in Section 5 in the main paper.

## E.1    Clustering method in the last step

In the last step of our approach, we use $k$-means to cluster the learned matrix $X_k$. We experimentally demonstrate here that the partition obtained is, in fact, independent of the clustering algorithm used in this step. Hence, in Figure 3, we compare spectral clustering with k-means. As in the main paper, we here consider varying the number of observations, $|\mathcal{T}|, |\mathcal{Q}|$ and varying the crowd noise $\epsilon$ for both the setting where $k$ is estimated by SPUR and where we consider $k$ to be known. There is no differences between the ARI obtained when using k-means or spectral clustering.

## E.2    Compare SPUR with known $k$

An important question is how good is SPUR at estimating the true number of clusters. We illustrate this in Figure 4. We start by comparing the first two columns, showing how the ARI changes for various parameters of the planted model. In the setting of $|\mathcal{Q}|, |\mathcal{T}| = n(\ln n)^3$ we see that using a known number of clusters outperforms SPUR, especially in parameter ranges that are harder to cluster (e.g. small $\delta, \epsilon$ or for a larger number of clusters). If we consider $|\mathcal{Q}|, |\mathcal{T}| = n(\ln n)^4$, SPUR correctly estimates the number of clusters and thus we omit the plots with known $k$.

## E.3    Experimental details for changing $F_{in}, F_{out}$ in the planted model

In this section, we give implementation details on the different distributions considered in Figure 1c. In the following let $\phi$ be the normal pdf and $\Phi$ the normal cdf. Recall that, in all the experiments, we fix $\delta = 0.5$ as the default.

**Parameters for $F_{in}$ and $F_{out}$ normal distributions.** Let $F_{in} = \mathcal{N}(\mu_{in}, \sigma)$ and $F_{out} = \mathcal{N}(\mu_{out}, \sigma)$. We fix $\sigma = 0.1$ and $\mu_{out} = 0$. Using $\delta$ we can compute $\mu_{in}$. Indeed, in this case, the cumulative distribution function is known and, thus, by setting it equal to $\mathbf{P}_{w \sim F_{in}, w' \sim F_{out}}(w > w') = \frac{1+\delta}{2}$ for some $\delta \in (0, 1]$ (as given in Equation (3)) we directly get the $\delta$ defined in Section 2: $\delta = 2\Phi\left((\mu_{in} - \mu_{out})/(\sqrt{2}\sigma)\right) - 1$. Then, assuming that $\mu_{out} = 0$, we get $\mu_{in} = \sqrt{2}\sigma\Phi^{-1}\left(\frac{1+\delta}{2}\right)$.

**Parameters for $F_{in}$ and $F_{out}$ Beta distributions.** Let $F_{in} = \text{Beta}(\alpha, \beta)$, $F_{out} = \text{Beta}(\alpha', \beta')$. We set $\alpha' = \beta' = 1$ such that $F_{out} = \text{Beta}(1, 1) = \text{Unif}(0, 1)$. We can then compute

$$\mathbf{P}_{w \sim \text{Beta}(\alpha, \beta), w' \sim \text{Beta}(1,1)}(w > w') = \mathbb{E}_w \left[ \int_0^w dw' \right]$$

$$= \mathbb{E}_w \left[ w \right]$$

$$= \frac{\alpha}{\alpha + \beta}$$

where the last line follows from the mean of the Beta distribution. Setting this equal to $\frac{1+\delta}{2}$ and solving for $\alpha$ gives: $\alpha = \beta \left( \frac{1+\delta}{1-\delta} \right)$. In our experiments, we fix $\beta = 2$.

**Parameters for $F_{in}$ Normal and $F_{out}$ Uniform.** Let $F_{in} = \mathcal{N}(\mu, 0)$, $F_{out} = \text{Unif}(0, 1)$. To set $\mu$, we compute:

$$\mathbf{P}_{w \sim \mathcal{N}(\mu, 0), w' \sim \text{Unif}(0,1)}(w > w') = \int_0^\infty \phi(w - \mu) dw \left[ \int_0^{\min(w, 1)} dw' \right]$$

$$= \int_0^1 w\phi(w - \mu)dw + \int_1^\infty \phi(w - \mu)dw + \mu \left( \Phi(1 - \mu) - \Phi(-\mu) \right)$$

$$= 1 + \phi(-\mu) - \phi(1 - \mu) + (\mu - 1)\Phi(1 - \mu) - \mu\Phi(-\mu)$$

Solving numerically for $\mu$ gives $\mu = \frac{1+\delta}{2}$.

### E.4 Influence of different planted model parameters

In this section we present additional experiments where we vary various parameters of the planted model. Recall that we consider the following parameters as default: $n = 1000$, $k = 4$, $\epsilon = 0.75$, $|\mathcal{T}| = |\mathcal{Q}| = n(\ln n)^4$ and $F_{in} = \mathcal{N} \left( \sqrt{2}\sigma\Phi^{-1}\left(\frac{1+\delta}{2}\right), \sigma^2 \right), F_{out} = \mathcal{N}\left(0, \sigma^2\right)$ with $\sigma = 0.1$ and $\delta = 0.5$.

**Number of samples $n$, first row in Figure 4.** We can first note that for $|\mathcal{Q}|, |\mathcal{T}| = n(\ln n)^3$ there is no difference in the behaviour between SPUR and known $k$. Both AddS-3 and AddS-4 achieve full recovery while MulK-3 and MulK-4 predictions are random. To learn somewhat meaningful partitions with MulK-3, one needs to increase the number of observations to $n(\ln n)^4$. However, even with this many comparisons, MulK-4 still learns random clusters.

**Intrinsic noise $\delta$, second row in Figure 4.** Using $|\mathcal{Q}|, |\mathcal{T}| = n(\ln n)^3$, we see that, for both SPUR and known $k$, AddS-3 and AddS-4 exactly recover the clusters even when the intrinsic noise is high, that is $\delta = 0.4$. MulK-3 and MulK-4 can only make random predictions in this case. When the number of observations increases to $n(\ln n)^4$, AddS-3 and AddS-4 exactly recover the clusters even for values of $\delta$ that are as small as $0.25$. In this case, MulK-4 still predicts random clusters, while MulK-3 is able to recover the clusters when the intrinsic noise is sufficiently small, that is $\delta \geq 0.6$.

**Crowd noise $\epsilon$, third row in Figure 4.** This parameter was already analyzed in the main paper. The plots are recalled here for the sake of completeness.

**Number of clusters $k$, fourth row in Figure 4.** Finally, we vary the number of planted clusters. Here, we observe the most noticeable difference between SPUR and known $k$. For $|\mathcal{Q}|, |\mathcal{T}| = n(\ln n)^3$, AddS-3 and AddS-4 with SPUR achieve perfect recovery for up to five clusters. While we notice a similar behaviour for AddS-3 and AddS-4 with known $k$, the drop in ARI only starts for $k > 7$ and is far less important than with SPUR. For $n(\ln n)^4$ observations AddS-3 and AddS-4 consistently recover all the clusters. On the other hand, MulK-3 only recovers clusters up to $k = 3$ (here, MulK-3 uses the number of clusters estimated by AddS-3 with SPUR, that is $k = 3$). Once again, MulK-4 can only make random predictions.

## F Further results for experiments on real comparison based data

In this final section we present supporting results for the real data experiments presented in Section 5.

Figure 4: Further experiments on the planted model. On the one hand, SPUR needs sufficiently many comparisons to correctly estimate the number of underlying clusters. On the other hand, our approaches are not overly sensitive to changes in the planted model parameters and are able to exactly recover the planted clusters with $n(\ln n)^3$ comparisons even in fairly difficult cases (small $\delta$, high $k$, ...). Furthermore, given $n(\ln n)^4$ comparisons, our approaches are able to exactly recover the planted clusters in all the considered cases.

Figure 5: ARI between the clustering obtained by the different baselines. AddS-3 and AddS-4 with SPUR both estimate that the number of cluster is $k = 2$. There is a high degree of agreement between the different approaches.

## F.1  Details on the Car dataset

The Car dataset (Kleindessner and von Luxburg, 2016) is a comparison based dataset that contains 60 examples grouped into 3 classes (SUV, city cars, sport cars) with 4 outliers. This dataset originally comes with a set of 6056 comparisons of the form "$x_i$ is most central in the triple $x_i$, $x_j$, $x_k$." Each of these comparisons corresponds to two triplets: "$x_j$ is more similar to $x_i$ than to $x_k$" and "$x_k$ is more similar to $x_i$ than to $x_j$." Hence, we have access to 12112 triplet comparisons.

## F.2  Food Dataset

In addition to the Car dataset we now look at a second comparison based dataset called Food (Wilber et al., 2014). It contains 100 food images and comes with 190376 triplet comparisons. Since there are no ground truth labels for the food dataset, we use the number of clusters estimated by SPUR for all methods and plot, in Figure 5, the similarity matrix between the different clustering approaches considered. Here, there is a high degree of agreement between all the clustering methods. Thus, most approaches predict the same clusters up to minor differences for a few data points. In Figure 7, we plot the clusters obtained by AddS-3 with SPUR (estimated $k$ is 2). The two clusters seem to separate *Sweet foods* from *Savoury foods*. Intuitively, it seems indeed natural for humans to judge that two sweet foods are more similar to each other than to a third savoury food.

## F.3  MNIST

In this section, we consider additional experiments on the MNIST dataset. First, we consider a second similarity measure to generate the triplets. Then, we illustrate the partitions obtained with AddS-3 with known $k$ and SPUR respectively.

**Gaussian similarity.** In the main paper, we use the Gaussian similarity to generate the comparisons. More precisely, we compute the similarity between two examples $x_i$ and $x_j$ as

$$w_{ij} = \exp\left(\frac{\|x_i - x_j\|_2^2}{\gamma^2}\right) \text{ with } \gamma = 1.$$

**Cosine similarity.** Instead of the Gaussian similarity, we could consider alternatives to generate the comparisons. For example, the Cosine similarity:

$$w_{ij} = \frac{\langle x_i, x_j \rangle}{\|x_i\|_2 \|x_j\|_2}.$$

In Figure 6, we show that using this alternative similarity affects the absolute results of the considered approaches. However, it does not change the overall trend, that is, as the number of comparisons increases, AddS-3 converges to the baseline of $k$-means with access to the original similarities.

**Clustering using known $k$.** Figure 8a shows the t-SNE embedding of 2000 MNIST samples of all ten classes, where we see a clear separation between some classes (for example, 0 and 1) and very close embedding between others (for example, 1 and 9). Note that the classes obtained by AddS-3 are shown up to permutations and may not reflect the majority label in the different clusters. Further note that the data presented here corresponds to a single repetition out of the 10 repetitions used to

(a) MNIST 1vs.7, $n = 2163$    (b) MNIST 10, $n = 2000$

Figure 6: Experiments on MNIST using the cosine similarity. The absolute ARI performances are different from the Gaussian similarity. However, the overall trend is preserved and, given sufficiently many comparisons, all the ordinal baselines reach the performance of $k$-means on the original data.

compute the mean ARI (with standard deviation) in the main paper and this appendix. In Figure 8d, we see that, for $|\mathcal{T}| = n(\ln n)^2$, the learned partition is not very representative of the original labels. Figure 8c shows that, when the number of comparisons increases to $|\mathcal{T}| = n(\ln n)^3$, the recovery ability of AddS-3 is greatly improved. However, the obtained partitions are not entirely satisfactory. Finally, Figure 8b shows that, when the number of comparisons further increases to $|\mathcal{T}| = n(\ln n)^4$, the clustering obtained is close to the true labeling and most clusters are correctly identified.

**Clustering using SPUR.** In this second set of experiments, we extend our observations from the previous paragraph to the labeling obtained by AddS-3 using SPUR. One can note that SPUR always underestimates the number of clusters. Hence, in Figure 9a, with $|\mathcal{T}| = n(\ln n)^3$, the number of predicted clusters is $k = 6$ while, in Figure 9b, with $|\mathcal{T}| = n(\ln n)^4$, the number of predicted clusters is $k = 8$. This explain the slightly worse behaviour of SPUR compared to known $k$ in Figure 2b in the main paper. Nevertheless, the difference in average ARI is not so significant when $|\mathcal{T}| = n(\ln n)^4$, suggesting that $8$ clusters is, in fact, a good estimate of the number of clusters that can reliably be distinguished by the different methods.

Class A with 36 images (*Sweet?*)

Class B with 64 images (*Savoury?*)

Figure 7: Clusters obtained by AddS-3 on the food dataset. It seems that the Sweet foods are separated from the Savoury ones.

(a) MNIST embedding with true labels

(b) AddS-3 $k = 10$, $|\mathcal{T}| = n(\ln n)^2$

(c) AddS-3 $k = 10$, $|\mathcal{T}| = n(\ln n)^3$

(d) AddS-3 $k = 10$, $|\mathcal{T}| = n(\ln n)^4$

Figure 8: t-SNE embedding of 2000 MNIST samples with (8a) true labeling and (8d)–(8b) clusters obtained by AddS-3 with known $k = 10$ and varying number of observations. The classes are given up to permutations and may not reflect the majority label in each cluster.

(a) AddS-3 SPUR, $|\mathcal{T}| = n(\ln n)^3$

(b) AddS-3 SPUR, $|\mathcal{T}| = n(\ln n)^4$

Figure 9: t-SNE embedding of 2000 MNIST samples with the clusters predicted by AddS-3 using SPUR and varying number of comparisons. The classes are given up to permutations and may not reflect the majority label in each cluster.