[Reviews · NeurIPS 2020]

Review 1

Summary and Contributions: The explicit similarities between the objects are not always available for clustering. This paper provides a method to cluster objects, when only the ordinal comparisons are available, such as “object i is more similar to object j than to k”. It estimates a pairwise similarity matrix from the comparisons before using a clustering method based on semi-definite programming (SDP). It also theoretically shows that its approach can exactly recover a planted clustering using a near-optimal number of passive comparisons. The theoretical finding is validated on real data. ---- Update --- The authors have adequately clarified the questions. After going through the rebuttal, this helps in improving my confidence score from 3 to 4.

Strengths: Considering a planted model for standard clustering , this paper shows that, when the number of clusters k is constant, Ω (n(ln n)^2) passive triplets or quadruplets are sufficient for exact recovery, which is comparable to the number of active comparisons (Ω (n (ln n))). This paper proposes the first analysis that considers and shows the impact of both the observation error and intrinsic noise on the number of passive comparisons. This paper proposes a new similarity function whose construction is more efficient than previous kernels. This paper proves a generic theoretical guarantee that holds for any similarity matrix and, thus that could be of interest beyond the comparison based setting.

Weaknesses: No

Correctness: No errors are found.

Clarity: This paper is well written in general.

Relation to Prior Work: The dicussion of prior work is adequate. The paper uses the SDP approach for clustering from similarity matrices by (Yan et al 2018; Chen and Yang 2020). The proof of Proposition 1 is adapted from Yan et al. (2018) , but proves the uniqueness of X* which was not done in the previous work. It requires significantly less passive comparisons comparing to previous work (Ghoshdastidar et al. 2019; Emamjomeh-Zadeh and Kempe, 2018). It discuss that most of previous analyses of SDP clustering either assume sub-Gaussian data (Yan and Sarkar, 2016) or consider similarity matrices with independence assumptions (Chen and Xu, 2014; Yan et al., 2018), while it is not required in this paper.

Reproducibility: Yes

Additional Feedback: It may be better to provide more information on the datasets (MNIST AND Food) used in the experiment section. It may be better to descrite the full algorithm proposed in the main text.


Review 2

Summary and Contributions: This paper studies comparison-based clustering from a planted-clustering perspective, developing methods that are nearly optimal when the data is generated by the planted clustering model, a generative model similar to the planted partition (aka SBM) random graph model. The work introduces two kernels, AddS-3 and AddS-4, that summarize the information contained in triplet or quadlet data (respectively) in an n x n matrix S. The paper then shows that when the triplet/quadlet data is generated under the model, the planted clustering is recovered. The solution approach, based on an SDP, borrows heavily from analogous existing methods for studying SBMs, but that is OK, this is a nice import to comparison-based clustering, and there is still significant technical novelty. See, for example, the uniqueness result in Proposition 1. The paper validates the efficiency of the methods on synthetic data, then on “semi-synthetic data” (MNIST data, with noise added to similarities from an embedding), then real data. The results on the real data are poor, but that may be amendable with more work to e.g. borrow from extensions in the SBM literature. Update: I thank the authors for a thoughtful response letter.

Strengths: + The paper is well written, with clear theorem statements, solid coverage of the literature, and well-documented empirics. + The synthetic, semi-synthetic, and real data empirics should be commended for a paper that is otherwise quite focused on theory. + The result is important/surprising, that at least under the planted model, the number of triplet/quadlet comparisons to achieve recovering of a planted clustering is less than expected.

Weaknesses: - The empirical results for real (car) data are quite weak, likely due to the planted clustering model being a poor fit to the data. But this could potentially be addressed by considering variations on the model that do a better job of modeling the data.

Correctness: They appear to be correct, though the paper is hard to fully digest without relying on the appendix, even omitting the proofs. Some further details could be provided on for the MNIST data (1v7 and full-10) on how the comparison triplets/quadlets were generated. I assume triplets/quadlets were selected uniformly at random, with replacement, using the Gaussian similarity described in the appendix. What happens if the triplets/quadlets aren't uniformly at random, but biased towards certain subsets? What happens if the "noise" is much more complicated than Gaussian, e.g. a mixture of populations that have different perceptions of the latent space?

Clarity: Yes.

Relation to Prior Work: Yes.

Reproducibility: Yes

Additional Feedback: The kernel representations AddS-3 and AddS-4 can be thought of, I think, as “all the information” when the data generating process is the block-based model in Section 2. If those are the true generating processes for triplets/quadlets, are these S then "sufficient statistic" of the model somehow? I don't know quite how to formalize that question, but there is some work on (minimal) sufficient statistics for the SBM that might help guide the authors here. It would make the use of AddS-3/4 much more intuitive. Further, I'll reiterate my main worry with the work, that the assumptions of the model are strong and the SDP algorithm fails when model is badly misspecified. Some thoughts on how robust different SBM-recovery algorithms are is covered in Moitra et al. (2016) "How Robust are Reconstruction Thresholds for Community Detection?", where SDP-based methods generally do well.


Review 3

Summary and Contributions: Usual clustering algorithms use similarity measures between pairs of data items. This paper considers the clustering problem when a similarity measure is not available; instead, only comparisons of the form "item i is more similar to item j than item k" (triples) or "items i and j are more similar to each other than items p and q" (quadruples) are available. It is also assumed that the clustering must be done in the "passive mode"; that is, the set of triples and quadruples is given in advance and that no new queries (for triple or quadruple comparisons) can be issued by an algorithm. The paper also considers the case where the comparisons are incorrect with a certain probability. (This can happen if the comparisons are generated by means such as crowd sourcing.) The paper uses the results of the comparisons to construct a similarity matrix for the data items and construct a clustering. It is shown that the number of triples/quadruples used to recover clusters under the planted cluster model is substantially smaller than previous work. An extensive set of experimental results are presented in the main paper and the supplement to validate the theoretical results. Update: This reviewer went through the other reviews and the author feedback. The author(s) has (have) adequately addressed all the points raised by the reviewers. This reviewer feels more confident about the quality of the paper. As a result, the reviewer has updated the confidence score from 3 to 4.

Strengths: -- Improved upper bounds on the number of comparisons needed under the passive comparison model. (Recall that the model assumes that the comparisons may be incorrect with a small probability.) The paper also provides a very good discussion of prior work and an extensive list of references. -- Non-trivial theoretical results that prove how a similarity matrix can be constructed from comparisons and a planted clustering can be recovered. -- The experimental results to validate the theoretical findings are comprehensive.

Weaknesses: There are two minor weaknesses. First, in many places, the paper uses the lower bound notation (i.e., the \Omega function) when presenting upper bounds. In a few places, there is also the opposite usage (i.e., upper bound notation is used when a lower bound is indicated). Some of these are pointed out in the "Additional feedback" section. In a few places, the authors need to slightly improve the presentation for readers to clearly understand the material.

Correctness: As far this reviewer can tell, the theoretical results seem correct and the experimental methodology seems sound.

Clarity: The paper is written very well. Only a few minor clarifications (indicated in the "Additional feedback" section) are needed.

Relation to Prior Work: The paper provides a very good discussion of prior work and clearly points out the new contributions relative to existing work.

Reproducibility: Yes

Additional Feedback: The topic of comparison-based clustering is attracting attention in the literature. This paper makes a very useful contribution to the topic by considering the passive comparisons model and obtaining better upper bounds on the number of comparisons used to carry out clustering. Overall, the paper is well written. It is interesting to see that for the Food data set, the experiments suggest the possibility that of an alternative natural clustering with two clusters instead of the assumed number of three clusters. The paper also discusses prior work in detail and clearly brings out the new contributions. I. Technical suggestions: (a) In many places, the paper uses the lower bound notation (i.e., the \Omega function) when presenting upper bounds. As an example, on page 2, line 65, the authors say "... \Omega(n (\ln{n})^2) passive triplets or quadruples are sufficient for exact recovery" when the intention is to say that O(n (\ln{n})^2) passive triples or quadruples are sufficient. (This type of error occurs on page 2, 5, 6 and 8.) In a few places, there is also the opposite usage; that is, upper bound notation is used when a lower bound is indicated. As an example, on page 2, line 88, the authors say "... require up to O(n) passes" where they want to convey that \Omega(n) passes are required. (b) Page 2, lines 68--69: The meaning of the sentence "Furthermore, it is near-optimal ... at least \Omega(n) comparisons" is not clear. (c) Page 3, Equation (1): Why do you say that you have accesses to triplets "or" quadruples? In general, your algorithm can use both triples and quadruples. (d) Page 3, lines 104--106: The reason for assumption that when a triple i, j, r is observed, either (i, j, r) \in T or (i, r, j) \in T is not clear. You also use a similar assumption about quadruples. (Since you are using the passive comparison mode, you have no control over which triples or quadruples appear in the input.) (e) Page 4, line 160: The use of \Sigma as a k by k matrix is likely to cause confusion because you are also using in terms of the form \Sigma_{\ell,\ell'} which looks like a summation rather than an entry in matrix. It is better to use a different symbol for the k by k matrix. (f) Page 5, line 199: Should the excess value of inter-cluster similarity be \Omega(p \epsilon \delta {n \choose 2}) instead of just p \epsilon \delta {n \choose 2}? (The \Omega seems necessary for consistency with the similar quantity for quadruples used in line 204 on the same page.) (g) Page 6, Remark 1: This remark also seems to need the assumption regarding similar cluster sizes. This should be checked. (h) Page 7, line 276: it would be better to indicate whether k_{\lambda} should be defined as the ceiling or floor function of trace(X_{\lambda}). (i) Page 7, last paragraph: You state the bound number of comparisons as "n (\ln{n})^{4.5}", etc. It is better to add a note earlier in the paper to indicate that such bounds refer to the ceiling of the indicated functions. II. Typos and stylistic suggestions: (a) Page 2, line 46: "exists" ---> "exist" (b) Page 2, line 85: "type" ---> "types" (c) Page 3, line 132: Insert "satisfy" after the word "means" (d) Page 3, line 135: "identify the number of" ---> "obtain bounds on" (e) Page 4, line 183: "makes" ---> "make" After author feedback: I thank the authors for the feedback. I have read through the other reviews and the author feedback. My overall score remains unchanged.

[Author Response · NeurIPS 2020]

We thank the reviewers for the positive recommendation for the paper, and also for the constructive feedback. Since, no critical concern has been raised, we list our answers for the different comments/suggestions made by the reviewers.

**Reviewer 1.**

**More information on the datasets used in the experiment section.** The original MNIST dataset is a classification dataset split into two parts for training and testing. For the purpose of clustering, we propose to use the test set. This dataset contains 10000 examples that are roughly equally distributed between the ten digits. In our first experiment (MNIST 1vs7), we select all the 1 and 7 (2163 examples in total). In the second experiment (MNIST 10), we subsample, without replacement, 2000 examples from the whole test set. In both case, we use the Gaussian similarity (See Section F.3 in the supplementary) to generate the comparisons and we randomly and uniformly draw, without replacement, between $n(\ln n)^2$ and $n(\ln n)^4$ comparisons to be observed by the different approaches. The food dataset contains 100 examples and 190376 triplet comparisons. Unfortunately, there is no ground truth and, thus, quantitatively assessing the quality of the obtained partitions is difficult. We will include this in the final version.

**Full algorithm in the main text.** We will utilise the additional page in camera ready version to add this.

**Reviewer 2.**

**The empirical results for real (car) data are quite weak** ... **could potentially be addressed by considering varia-tions on the model. + Robust reconstruction (Moitra et al)** The suggestion of considering semirandom models to address mis-specification is very helpful. We will consider (and mention) this as part of future work. Furthermore, we note that the comparisons were obtained from a crowd with view different from the expert who labelled the cars. This would call for modelling noise in crowdsourcing. For example, rather than the type of cars, the crowd might focus on its color. This might explain the differences between our partition based on comparisons and the ground truth. This might also explain why SPUR only detected 2 clusters instead of 3 (as noted by Reviewer 3).

**Further details could be provided on for the MNIST data (1v7 and full-10) on how the comparison triplets/quadlets were generated.** Please see our answer to the first question of Reviewer 1.

**AddS-3/4 kernels as "sufficient statistic" for the model.** We thank the reviewer for this suggestion. As noted in the review, this statement is difficult to formalise. We will think about this and, if possible, add it in the follow-up work.

**Reviewer 3.**

**a) Paper uses the lower bound notation (i.e., the $\Omega$ function) when presenting upper bounds.** We thank the reviewer for this comment and we agree that some of our notations were not correct. We will check all of them and make the appropriate changes.

**b) Meaning of lines 68-69 not clear.** In order to accurately select which partition an item $x$ belongs to, we need to have some information about it. Otherwise, $x$ can only be assigned to a random cluster. Since, in our setting, we only have access to the examples through comparisons, we need to observe at least one comparison for each example. Hence, $\Omega(n)$ comparisons is a necessary requirement to cluster $n$ items. We will clarify this in the final version.

**c) In general, your algorithm can use both triples and quadruples.** Yes, but this usually does not happen as in practice a single type of comparison is obtained from the crowd (typically triplets). In fact, we are not aware of any real world scenario where multiple types of comparisons are collected simultaneously.

**d) lines 104-106 not clear in context of passive comparisons.** These sentences have only a mild underlying as-sumption, that is, all $w_{ij}$ are distinct (this is not strictly necessary, but it slightly simplifies the analysis). Assuming distinctness, there are no further assumptions. If the triple $i, j, r$ with $i$ as reference *is observed*, then obviously either $w_{ij} > w_{ir}$ or $w_{ij} < w_{ir}$. If the reviewer meant that both could occur when we have multiple observations of the same comparison. We note that this is rare in practice but, in the experiments, we simply follow Ghoshdastidar et al (2019) and use a majority vote.

**f) Should the excess value of inter-cluster similarity be $\Omega(p\epsilon\delta\binom{n}{2})$ instead of just $p\epsilon\delta\binom{n}{2}$?** The value $p\epsilon\delta\binom{n}{2}$ for quadruplets is exact. In fact, Equation (21) in Appendix C.2 also gives the exact value for the triplet setting as $p\epsilon\delta(n-2)$. We will remove the unnecessary $\Omega(\cdot)$.

**g) Remark 1 also seems to need the assumption regarding similar cluster sizes.** Yes, all the remarks are stated assuming similar cluster sizes. We quickly mention this in the last paragraph on page 5. We will make this more explicit.

**h) it would be better to indicate whether $k_\lambda$ should be defined as the ceiling or floor function of trace$(X_\lambda)$.** $k_\lambda$ is the rounded function of trace$(X_\lambda)$, that is the closest integer.

**Other textual comments.** Thank you, we will address them in the camera ready version.

[Meta-Review · NeurIPS 2020]

The paper shows how using only O(n\ln^2{n}) triplet queries (point i is more similar to j than k), it is possible to recover the exact clusters of a planted model. While the technique of using SDP for recovering planted clusters is not new, overall, the adaptation to handle comparison queries is quite nontrivial. The paper also provides empirical results on real and synthetic data.